# REASONING SELF-EVALUATION VIA TRAJECTORY DYNAMICS MODELING

## ABSTRACT

Large Reasoning Models (LRMs) enhance the reasoning ability of LLMs by explicitly generating chain-of-thought (CoT) trajectories before producing answers. Yet, how to enable LRMs to self-evaluate the correctness of their outputs remains largely unexplored, posing challenges for reliability. Existing approaches often rely on external supervision—such as ground-truth labels, auxiliary tools, or additional training—which largely reduces their applicability in real-world scenarios. In this work, we show that reasoning trajectories revealing the intermediate dynamics can be leveraged for self-evaluation, allowing models to self-assess the reliability of the reasoning without external supervision. We propose the $D^3$ Reasoning Score, a trajectory-based metric that quantifies output correctness built with three complementary components: Distance ($D_1$), Dynamism ($D_2$), and Difficulty ($D_3$), each of which can be derived purely based on internal signals of the model. Jointly modeling these factors yields a single scalar score aligned with output reliability, providing a fine-grained, white-box measure of reasoning fidelity. Extensive experiments across eight open-source LRMs and seven widely used benchmarks—covering both language and multimodal tasks—demonstrate that $D^3$ consistently outperforms prior baselines in AUROC, AUPR, and FPR@95. These results establish $D^3$ as a principled and practical metric for systematic self-evaluation and diagnosis of reasoning trajectories. The code is available for anonymous access via this repository.

## 1 INTRODUCTION

Large Reasoning Models (LRMs) DeepSeek-AI (2025); OpenAI (2024); Team (2025b) have achieved notable progress on complex reasoning tasks Chen et al. (2023); Cobbe et al. (2021); Hendrycks et al. (2021); Shi et al. (2022); Wang et al. (2024); Talmor et al. (2019) by generating answers through structured and deliberate reasoning trajectories. However, while such trajectories sometime are capable of yielding correct answers, they may also culminate in incorrect or even harmful outputs, raising concerns about the reliability of model predictions.

To address this issue, various approaches have been proposed to evaluate LRM reliability, which can be broadly grouped into two categories. The first relies on external assessment, leveraging human annotations, linear probing, or auxiliary judge models (Wang & Xu, 2025; Li et al., 2023; He et al., 2025). The second focuses on self-evaluation, which estimates correctness based solely on the model's own internal signals. Self-evaluation methods are further divided into white-box and black-box approaches: white-box methods quantify internal uncertainty measures—such as softmax confidence, temperature scaling, or hidden-state entropy—without external supervision (Wang et al., 2025b; Shih et al., 2023; Huang et al., 2025; Si et al., 2023), whereas black-box methods directly prompt the model to output confidence scores over its responses Sun et al. (2025). In particular, self-evaluation has emerged as a crucial direction for enhancing the reliability of LRMs. Its goal is to determine whether a model's answer to a given query is correct based solely on internal signals, without access to ground-truth labels, external tools, or additional supervised training. While effective in controlled, supervised settings, these external assessment methods lack generality and are impractical at deployment: they depend on external resources that may be unavailable and degrade significantly under distribution shift between training and real-world queries.

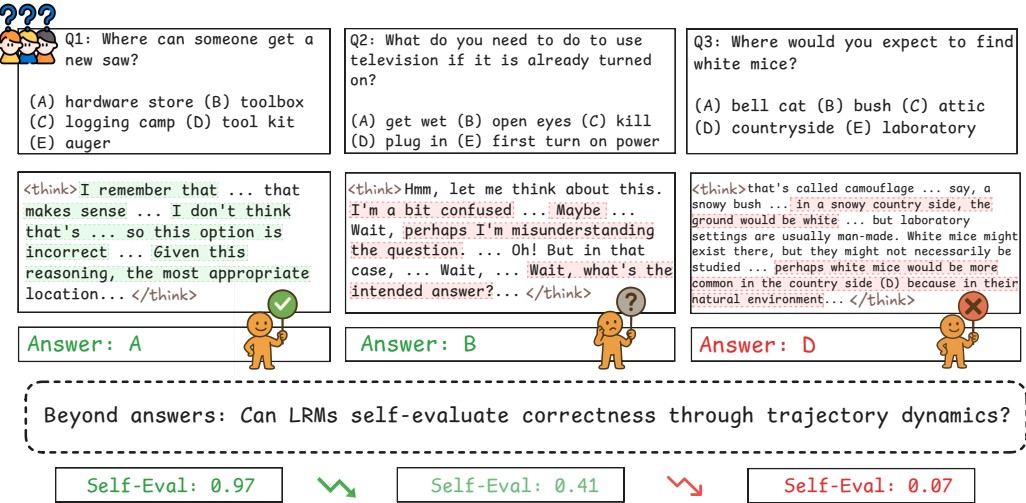

Figure 1: Illustration of self-evaluation through trajectory dynamics. For each query, an LRM produces both a final answer and a CoT reasoning trajectory. By analyzing hidden-state dynamics, the model assigns a self-evaluation score: higher scores correspond to coherent and stable reasoning trajectories with greater likelihood of correctness, while lower scores reveal unstable logic or flawed assumptions that correlate with incorrect answers.

Self-evaluation methods seek to overcome these limitations but still face critical challenges. Black-box approaches, which prompt models to output confidence scores, are systematically over-confident and offer limited interpretability (Sun et al., 2025; Wang et al., 2025b). White-box approaches instead probe internal signals such as softmax confidence, temperature scaling, or hidden-state entropy (Shih et al., 2023; Huang et al., 2025; Si et al., 2023), yet they often collapse the entire reasoning process into a single-step output distribution, leaving the rich trajectory $z_{0:T}$ underutilized. A more recent line of work explores hidden-state dynamics and introduces two separate CoE metrics to distinguish correct from incorrect trajectories (Wang et al., 2025b). However, these methods still lack a synthetic modeling of trajectory dynamics and a difficulty-aware calibration across queries, resulting in limited robustness and accuracy.

In this paper, we propose a novel white-box self-evaluation method based on Difficulty-Calibrated Trajectory Dynamics, termed the $D^3$-Reasoning Score. The central idea is to exploit the hidden-state trajectory generated during reasoning and translate its dynamics into a scalar measure that aligns with output correctness. $D^3$ integrates multiple complementary signals—capturing how reasoning progresses, how it adapts, and how query difficulty shapes reliability—into a unified score. Each component plays a distinct role in shaping the final evaluation, and together they provide a fine-grained, interpretable measure of reasoning fidelity. Detailed formulations and empirical demonstrations of these components will be presented in the following sections.

This design overcomes the limitations of externally supervised methods, which depend on annotated information. More importantly, compared with prior self-evaluation approaches—such as black-box prompting methods that directly elicit confidence scores or white-box uncertainty estimators based on single-step output distributions—$D^3$ seamlessly integrates trajectory dynamics from hidden-state evolution with query difficulty calibration. The three components—Distance, Dynamism, and Difficulty—are coherently unified within a single evaluation framework, substantially boosting LRM self-evaluation capability in a white-box manner.

We conduct comprehensive evaluations to assess the effectiveness of the proposed $D^3$-Reasoning Score. Three standard metrics—AUROC, AUPR, and FPR@95—are employed to measure how accurately $D^3$ reflects the correctness of reasoning trajectories. Experiments span eight open-source LRMs, including both language and multimodal models DeepSeek-AI (2025); OpenAI (2024); Team (2025b), and seven diverse benchmarks Chen et al. (2023); Cobbe et al. (2021); Hendrycks et al. (2021); Shi et al. (2022); Wang et al. (2024); Talmor et al. (2019); Lu et al. (2021) covering mathematics, commonsense reasoning, theorem proving, general knowledge retrieval, multilingual

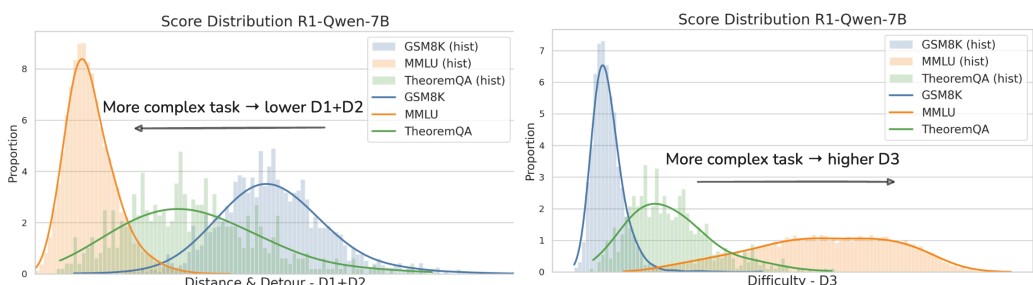

Figure 2: Score distributions of trajectory dynamics and difficulty across datasets of varying complexity (R1-Qwen-7B). Left: Distribution of $D_1 + D_2$, which reflects effective progress and adaptive trajectory dynamics. Simpler datasets (e.g., GSM8K) exhibit higher $D_1 + D_2$ scores, indicating that the model can more effectively explore and advance reasoning trajectories. Right: Distribution of $D_3$, the entropy-based difficulty calibrator. Here, $D_3$ grows consistently with dataset complexity, showing strong correlation with task difficulty. Together, these results validate $D_3$ as a difficulty-aware factor for calibrating trajectory dynamics in $D^3$.

understanding, and geometry problem solving. Empirical results demonstrate that $D^3$ consistently outperforms existing baselines across all settings, highlighting its robustness and generality as a trajectory-based self-evaluation metric.

Our contributions are summarized as follows:

- We identify that first-order differences of hidden states ($D_1$) capture effective progress, while second-order differences ($D_2$) capture adaptive trajectory dynamics. Furthermore, the entropy of the final predictive distribution ($D_3$) reflects query difficulty. By integrating these components, we propose the $D^3$-Reasoning Score. Building on this formulation, we develop a novel white-box self-evaluation method for LRMs. $D^3$ provides a unified scalar score that aligns with output reliability, where higher values indicate greater likelihood of correctness and lower values signal incorrectness.

- We conduct extensive experiments across eight open-source LRMs, spanning both language and multimodal models, and seven challenging benchmarks covering diverse domains and difficulty levels. Results consistently demonstrate that $D^3$ surpasses state-of-the-art baselines, establishing it as a more principled and effective modeling approach for reasoning self-evaluation.

## 2 SELF-EVALUATION VIA TRAJECTORY DYNAMICS MODELING

### 2.1 FORMULATION OF LRM CoT REASONING

LRMs unfold reasoning as a structured and deliberative process, where the model generates not only a conclusive answer but also a reasoning trajectory together as the outcome of inference, as the examples shown in Figure 1. Upon receiving a query task, an LRM will extract and encode the information contained in the question into a unified representation to facilitate the subsequent answering with thinking processes. Following the query embedding processing, the model starts its reasoning phase, where it generates intermediate reasoning tokens between designated special tokens such as `<think>` and `</think>`, which will be eventually translated into textual information as the reasoning trajectory.

For this specific reasoning phase, we assume the total sequence length of generated CoT reasoning tokens is $K$, wherein the $k$-th generated reasoning token is denoted as $q_k$. We further formalize the layer-wise inference process in the transformer for generating $q_k$ as follows. At each transformer layer, the hidden state of each token is updated based on the model parameters and the accumulated processing results from the previous layers. Formally, let $q_k^i \in \mathbb{R}^d$ denote the hidden state of the $k$-th token modified after the $i$-th layer computation, where $d$ is the hidden dimension. The state transition

from $q_k^{i-1}$ to $q_k^i$ can be represented as a transformation function below:

$$q_k^i = F_i(q_k^{i-1}, W^i), \ \forall i = 1, 2, \cdots, L, \tag{1}$$

where $W^i$ represents the synthesized effect of the model parameters and attention mechanisms applied to the $q_k^{i-1}$; and $L$ is the total number of transformer layers of an LRM.

Once the final reasoning token $q_K^L$ is produced, the reasoning phase concludes and the model transitions to generating the final answer conditioned on the trajectory. In this work, we propose to profile the entire reasoning trajectory and derive the $D^3$-Reasoning Score from its dynamics, enabling reliable self-evaluation of model outputs. We first apply mean pooling over the collected hidden states that are modified by the same transformer layer $i$ across all reasoning tokens. Specifically, we average all $K$ reasoning tokens generated by layer $i$, i.e., $\bar{q}^i = \frac{1}{K} \sum_{k=1}^{K} q_k^i$, where $\bar{q}^i$ serves as a compact semantic summary of the reasoning trajectory at the $i$-th layer, aggregating the representational states of all intermediate reasoning steps excluding the final answer tokens.

The main purpose of doing the mean pooling above is explained as follows. First of all, since these tokens are generated within the dedicated reasoning phase (i.e., between `<think>` and `</think>`), their averaged representation preserves the core semantics without information loss. Moreover, mean pooling operation reduces token-level variance and highlights the global progression of reasoning across layers, which is critical for modeling the layer-wise dynamics. Note that this simplification strategy has been widely adopted in prior work for semantic representation modeling Gao et al. (2021); Reimers & Gurevych (2019); Bi et al. (2025).

## 2.2 $D^3$-Reasoning Score: Distance · Dynamism · Difficulty

### 2.2.1 Trajectory Dynamics from Hidden State Evolution

Recent studies suggest that the hidden states of LRMs encode rich internal signals that strongly correlate with response correctness (Azaria & Mitchell, 2023; Liu et al., 2023). Unlike prior approaches that require correctness labels to train probing classifiers for extracting hidden-state features (Su et al., 2024; Burns et al., 2024), we instead analyze the dynamics of hidden-state evolution during reasoning. Specifically, we introduce two complementary components—Distance ($D_1$) and Dynamism ($D_2$)—to jointly quantify trajectory dynamics.

These components are computed from the hidden states of reasoning tokens generated in the reasoning phase. Formally, we collect the hidden representations into a $(K, L, d)$ tensor, where $K$ denotes the number of reasoning tokens, $L$ the number of transformer layers, and $d$ the hidden dimension. The trajectory dynamics are defined as:

$$D_1^i = \|\bar{q}^i - \bar{q}^{i-1}\|, \qquad D_2^i = \|(\bar{q}^i - \bar{q}^{i-1}) - (\bar{q}^{i-1} - \bar{q}^{i-2})\|, \tag{2}$$

where $\bar{q}^i$ is the layer-averaged hidden state at reasoning step $i$. Intuitively, $D_1$ measures first-order differences that capture effective progress, while $D_2$ measures second-order differences that capture exploratory adjustments—variable step sizes and direction shifts—reflecting adaptive search behavior.

In Figure 2, we visualize the distribution of $D_1 + D_2$ across tasks of different complexity. We consider GSM8K (simple mathematical QA), TheoremQA (hard mathematical reasoning), and MMLU (massive knowledge QA). The results reveal a clear trend: simpler tasks such as GSM8K yield higher $D_1 + D_2$ scores, whereas more complex tasks such as TheoremQA and MMLU yield lower scores. This indicates that when LRMs encounter queries with a higher likelihood of producing incorrect answers, their reasoning trajectories become flatter and less dynamic, revealing limited effective progress and weaker adaptive search capability. Crucially, this property provides an intrinsic signal that can be exploited for white-box self-evaluation, allowing LRMs to assess the reliability of their own reasoning without external supervision.

### 2.2.2 Entropy-based query difficulty calibration

To further enhance self-evaluation, we instantiate Difficulty ($D_3$) as the predictive entropy of the reasoning trajectory, defined as

$$D_3 = \mathbb{E}_{1 \le k \le K}\big[-\log p_k\big], \tag{3}$$

where $p_k$ denotes the probability assigned to the ground-truth token at reasoning step $k$. Intuitively, $D_3$ captures the average distributional uncertainty accumulated along the trajectory. Higher values of $D_3$ indicate greater uncertainty, reflecting harder queries. Empirically, $D_3$ increases consistently with dataset complexity (Figure 2, right), validating its role as a principled proxy for query difficulty.

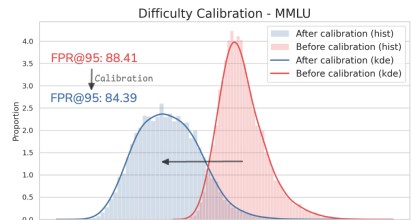

Crucially, the distributional behavior of $D_3$ complements the trajectory dynamics components $D_1 + D_2$ (Figure 2, left). For example, datasets where $D_1 + D_2$ values concentrate (e.g., MMLU) exhibit broader variability in $D_3$, reflecting a mixture of easy and hard queries. Without conditioning on $D_3$, queries of heterogeneous difficulty may collapse to similar $D_1 + D_2$ scores, which inflates false positives under high recall (i.e., elevated FPR@95).

Figure 3: Effect of difficulty calibration on MMLU. Incorporating $D_3$ smooths the score distribution and reduces FPR@95, improving the reliability of $D^3$ self-evaluation.

To mitigate this, we calibrate $D_1 + D_2$ with $D_3$, defining the final reasoning score $D^3$ as

$$
\begin{aligned}
D^3 &= D_1 + D_2 - D_3 \\
&= \sum_{i=2}^{L} \|\bar{q}^i - \bar{q}^{i-1}\| + \sum_{i=3}^{L} \|(\bar{q}^i - \bar{q}^{i-1}) - (\bar{q}^{i-1} - \bar{q}^{i-2})\| - \mathbb{E}_{1 \le k \le K}\big[ -\log p_k \big].
\end{aligned}
\tag{4}
$$

where $\bar{q}^i$ denotes the layer-averaged hidden state. This calibration smooths the score distribution and separates truly reliable trajectories from those associated with high difficulty but strong dynamics, thereby reducing FPR@95 and enhancing the robustness of self-evaluation (Figure 3).

## 3 Experiments

We first describe the experimental setup, including models, datasets, and evaluation metrics. We then compare the proposed $D^3$-Reasoning Score against strong baselines to assess its effectiveness as a self-evaluation method. Beyond overall comparison, we conduct in-depth analyses along several dimensions: (i) domain-specific versus cross-domain generalization, (ii) scaling behavior across different model sizes, and (iii) robustness under multilingual and multimodal reasoning scenarios. Finally, we perform ablation studies to examine the contribution of each design choice. In particular, we evaluate alternative strategies for reasoning-token aggregation, and disentangle the impact of the three components—Distance ($D_1$), Dynamism ($D_2$), and Difficulty ($D_3$)—on the overall performance of $D^3$. This systematic evaluation provides a comprehensive understanding of the proposed metric's reliability and generality.

### 3.1 Experimental Setup

**Assessment Criteria.** The central question is whether difficulty-calibrated trajectory dynamics provide a reliable basis for self-evaluation. Intuitively, the $D^3$ score should reflect the correctness of model outputs: trajectories with high $D^3$ should correspond to correct answers with high probability, while low $D^3$ should indicate a high likelihood of incorrectness. Conversely, if $D^3$ fails to align with correctness—e.g., assigning high scores to wrong answers—then the modeling of trajectory dynamics is unsatisfactory. Evaluating this alignment can be cast as a ranking problem. As illustrated in Figure 4 (left), the fewer incorrect samples (red) that appear in the high-score region, the stronger the self-evaluation capability. Technically, this alignment can be quantified by standard evaluation metrics. In this paper, we adopt three widely used metrics—AUROC, AUPR, and FPR@95—to measure how well $D^3$ satisfies the self-evaluation criterion.

  **AUROC:** captures global ranking quality. A higher AUROC indicates that the $D^3$-Reasoning Score consistently assigns higher values to trajectories leading to correct answers, reflecting strong overall discriminative power.

  **AUPR:** emphasizes precision under class imbalance. It measures whether $D^3$ can effectively prioritize correct trajectories when correct cases are relatively sparse. A higher AUPR reflects stronger ability to accurately isolate reliable reasoning in imbalanced settings.

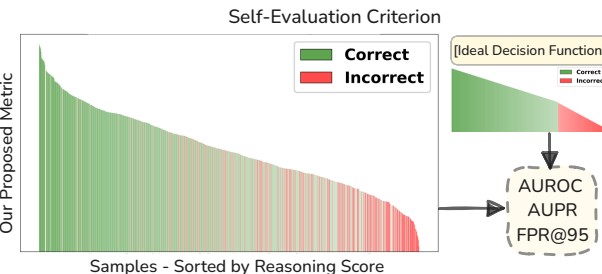 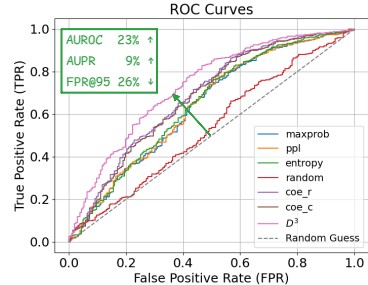

Figure 4: Left: Samples ranked by $D^3$ score, where higher scores align with correct outputs, reflecting the soundness of reasoning and logical causal relationship in terms of AUROC, AUPR, and FPR@95 metric. Right: ROC curves comparing $D^3$ score with baselines, where the dashed line denotes the expected performance of a random scoring function as a lower bound.

**FPR@95:** evaluates robustness under high recall. It reports the false positive rate when 95% of correct trajectories are recalled. Lower FPR@95 indicates that $D^3$ achieves reliable self-evaluation with minimal overestimation of incorrect trajectories, even under stringent correctness requirements.

These metrics are widely adopted for ranking-based binary classification, particularly under imbalanced or high-recall conditions. Together, they provide a rigorous and multi-faceted evaluation of how faithfully the $D^3$-Reasoning Score reflects the correctness of model outputs. Formal definitions are provided in Appendix A.7.

**Baselines.** We compare against seven baselines covering both white-box and black-box methods: `Random`, `Maximum Softmax Probability (MSP)` Wang et al. (2025b), `Perplexity` Si et al. (2023), `Entropy` Huang et al. (2025), `Verbal Self-Evaluation` Li et al. (2024), and two recent representation-based metrics, `CoE-R` and `CoE-C` Wang et al. (2025b). For all methods, we compute the three evaluation metrics introduced above. In particular, the `Random` baseline serves as a lower bound, corresponding to a uniform ranking with no self-evaluation capability.

**Models and Benchmarks.** We evaluate the proposed $D^3$-Reasoning Score on seven state-of-the-art LRMs spanning a wide range of scales: `DeepSeek-R1-Distill-Qwen-1.5B/7B/14B/32B`, `DeepSeek-R1-Distill-Llama-8B/70B` DeepSeek-AI (2025), and `QwQ-32B` Team (2025b). In addition, we conduct experiments on `Qwen2.5-VL` Bai et al. (2025) trained with GRPO to evaluate multimodal reasoning scenarios. Together, these models span diverse scales and modalities, offering a comprehensive testbed for examining the generality and robustness of $D^3$. Detailed model architectures and training configurations are provided in Appendix A.6.

To assess generality, we experiment on seven widely adopted benchmarks. MMLU and MMLU-Pro cover multi-domain reasoning across a broad range of subjects. GSM8K, TheoremQA, and CommonsenseQA probe distinct reasoning skills, including arithmetic problem solving, formal theorem reasoning, and commonsense understanding. We further evaluate cross-lingual robustness on MGSM, considering English, German, French, and Chinese subsets. Finally, we test multimodal reasoning with Geometry3K Lu et al. (2021), which requires spatial understanding from both textual and diagrammatic inputs. Comprehensive dataset details, including task formulations, prompts, and examples, are provided in Appendix A.5.

## 3.2 MAIN RESULTS

We present a comprehensive evaluation of the proposed $D^3$-Reasoning Score across multiple experimental settings. First, we examine its generality across diverse reasoning domains and task difficulties, including mathematical reasoning, commonsense reasoning, and multi-task knowledge reasoning, with results reported in Table 1 and Table 2. Next, Table 3 evaluates larger-scale models, including two 32B models and one 70B model, demonstrating the scalability of $D^3$ to stronger LRM architectures. We further assess multilingual robustness using MGSM subsets (English, German,

Table 1: Main results on GSM8K, CommonsenseQA, and TheoremQA. Arrows indicate direction of improvement.

| Model | Method | GSM8K | | | CommonsenseQA | | | TheoremQA | | |
|---|---|---|---|---|---|---|---|---|---|---|
| | | AUROC ↑ | AUPR ↑ | FPR@95 ↓ | AUROC ↑ | AUPR ↑ | FPR@95 ↓ | AUROC ↑ | AUPR ↑ | FPR@95 ↓ |
| R1-Qwen-1.5B | Random | 50.23 | 75.39 | 94.29 | 50.03 | 46.81 | 93.14 | 49.75 | 40.35 | 92.81 |
| | Verbal | 51.48 | 75.11 | 96.34 | 52.01 | 45.87 | 91.81 | 51.51 | 39.94 | 93.22 |
| | MaxProb | 70.34 | 87.97 | 76.95 | 64.06 | 61.16 | 81.42 | 53.65 | 43.91 | 92.19 |
| | PPL | 70.88 | 88.24 | 77.29 | 63.86 | 61.07 | 83.19 | 54.05 | 44.24 | 91.25 |
| | Entropy | 71.86 | 88.54 | 75.93 | 64.26 | 61.32 | 82.30 | 54.33 | 44.52 | 92.19 |
| | CoE-R | 74.17 | 89.99 | 75.93 | 60.03 | 56.37 | 85.18 | 53.33 | 44.18 | 93.44 |
| | CoE-C | 73.32 | 89.37 | 75.25 | 58.90 | 55.44 | 85.18 | 52.42 | 43.34 | 90.31 |
| | $D^3$ | **77.28** | **91.42** | **72.20** | **64.26** | **61.34** | **81.19** | **54.37** | **44.82** | **90.00** |
| R1-Qwen-7B | Random | 53.56 | 85.88 | 93.20 | 48.51 | 71.16 | 98.02 | 46.39 | 48.46 | 98.12 |
| | Verbal | 52.77 | 85.42 | 93.71 | 51.07 | 72.32 | 97.34 | 49.53 | 49.41 | 97.52 |
| | MaxProb | 67.05 | 92.26 | 82.42 | 76.64 | 86.89 | 67.46 | 48.95 | 49.88 | 89.85 |
| | PPL | 67.27 | 92.32 | 82.42 | 76.41 | 88.33 | 67.46 | 49.38 | 50.16 | 91.35 |
| | Entropy | 68.04 | 92.54 | 79.39 | 74.72 | 88.03 | 68.25 | 49.58 | 50.31 | 90.98 |
| | CoE-R | 70.95 | 92.98 | 73.94 | 71.91 | 85.58 | 75.79 | 50.99 | 53.31 | 90.60 |
| | CoE-C | 70.12 | 92.74 | 74.55 | 70.42 | 84.81 | 76.98 | 50.56 | 52.94 | 90.23 |
| | $D^3$ | **76.32** | **94.73** | **67.27** | **77.41** | **88.51** | **65.87** | 50.65 | 52.49 | **90.23** |
| R1-LLaMA-8B | Random | 50.22 | 81.21 | 95.33 | 51.75 | 77.22 | 96.14 | 46.29 | 41.67 | 92.91 |
| | Verbal | 51.22 | 85.49 | 94.77 | 51.04 | 75.95 | 97.41 | 49.11 | 40.24 | 93.57 |
| | MaxProb | 58.85 | 85.46 | 94.49 | 74.71 | 89.29 | 67.38 | 59.39 | 53.12 | 84.65 |
| | PPL | 58.94 | 85.51 | 94.49 | 74.55 | 89.23 | 67.81 | 60.01 | 53.91 | 85.04 |
| | Entropy | 60.25 | 85.89 | 94.88 | 74.68 | 89.28 | 65.24 | 60.27 | 54.06 | 84.65 |
| | CoE-R | 63.01 | 86.57 | 86.22 | 52.00 | 77.92 | 93.56 | 50.56 | 46.02 | 96.46 |
| | CoE-C | 63.86 | 86.50 | 86.61 | 73.90 | 88.92 | 69.96 | 58.01 | 54.10 | 93.70 |
| | $D^3$ | **68.12** | **89.37** | **80.31** | **74.71** | **89.35** | **65.24** | **60.57** | **56.06** | **83.86** |
| R1-Qwen-14B | Random | 49.46 | 93.23 | 95.24 | 49.13 | 84.30 | 95.03 | 50.93 | 56.79 | 94.93 |
| | Verbal | 50.44 | 91.91 | 96.14 | 54.19 | 85.87 | 94.85 | 51.83 | 56.54 | 94.72 |
| | MaxProb | 53.98 | 93.06 | 85.71 | 78.90 | 94.87 | 65.22 | 51.94 | 57.71 | 89.40 |
| | PPL | 54.51 | 93.23 | 84.52 | 79.24 | 94.99 | 64.60 | 52.39 | 58.07 | 89.40 |
| | Entropy | 55.13 | 93.26 | 85.71 | 79.24 | 94.97 | 65.84 | 52.62 | 58.24 | 90.32 |
| | CoE-R | 59.62 | 93.84 | 77.38 | 71.78 | 91.82 | 73.29 | 52.28 | 58.32 | 93.55 |
| | CoE-C | 58.63 | 93.66 | 77.38 | 68.40 | 90.78 | 80.75 | 50.99 | 57.39 | 92.63 |
| | $D^3$ | **61.64** | **94.04** | **76.19** | **79.27** | **95.00** | **61.49** | **52.65** | **58.77** | **88.02** |

Table 2: Performance comparison on MMLU and MMLU-Pro, two multi-task benchmarks requiring both world knowledge and complex reasoning.

| Dataset | Method | R1-Qwen-1.5B | | | R1-Qwen-7B | | | R1-LLaMA-8B | | | R1-Qwen-14B | | |
|---|---|---|---|---|---|---|---|---|---|---|---|---|---|
| | | AUROC ↑ | AUPR ↑ | FPR@95 ↓ | AUROC ↑ | AUPR ↑ | FPR@95 ↓ | AUROC ↑ | AUPR ↑ | FPR@95 ↓ | AUROC ↑ | AUPR ↑ | FPR@95 ↓ |
| MMLU-Pro | Random | 49.51 | 25.83 | 95.72 | 50.71 | 30.83 | 94.96 | 50.95 | 31.25 | 94.30 | 50.22 | 40.12 | 94.31 |
| | MaxProb | 61.44 | 33.40 | 91.05 | 60.11 | 35.55 | 91.87 | 59.29 | 36.68 | 90.54 | 65.54 | 48.33 | 85.52 |
| | PPL | 61.65 | 33.38 | 90.89 | 60.31 | 35.86 | 90.96 | 59.42 | 36.86 | 90.44 | 65.34 | 47.98 | 86.23 |
| | Entropy | 62.28 | 33.93 | 90.57 | 59.89 | 34.89 | 90.10 | 59.82 | 37.12 | 89.93 | 66.12 | 48.31 | 85.45 |
| | CoE-R | 57.78 | 30.76 | 91.25 | 52.63 | 31.34 | 92.81 | 51.77 | 33.66 | 94.12 | 61.45 | 45.12 | 86.77 |
| | CoE-C | 54.50 | 28.45 | 92.95 | 55.41 | 32.89 | 92.13 | 55.41 | 36.20 | 89.60 | 61.52 | 46.53 | 87.10 |
| | $D^3$ | **62.46** | 33.39 | **90.41** | **61.89** | **36.13** | **89.45** | **60.47** | **37.90** | **89.24** | 65.89 | **48.57** | **85.12** |
| MMLU | Random | 50.22 | 57.23 | 95.80 | 48.27 | 70.83 | 95.42 | 50.29 | 78.96 | 95.75 | 50.65 | 85.72 | 94.69 |
| | MaxProb | 64.61 | 75.37 | 91.81 | 72.11 | 87.04 | 84.77 | 71.29 | 89.55 | 82.54 | 78.20 | 93.77 | 71.69 |
| | PPL | 65.19 | 75.50 | 90.96 | 72.33 | 87.22 | 84.36 | 71.32 | 89.68 | 83.25 | 78.26 | 94.03 | 72.28 |
| | Entropy | 66.40 | 76.38 | 89.95 | 73.23 | 87.65 | 83.62 | 71.87 | 89.87 | 83.16 | 78.17 | 93.73 | 72.42 |
| | CoE-R | 58.79 | 65.18 | 93.20 | 64.94 | 80.61 | 85.79 | 49.49 | 80.25 | 95.31 | 71.59 | 92.94 | 82.95 |
| | CoE-C | 56.34 | 62.21 | 93.97 | 62.74 | 79.32 | 88.27 | 66.34 | 86.60 | 85.83 | 69.34 | 92.39 | 87.02 |
| | $D^3$ | **68.14** | **76.47** | 90.36 | **74.26** | **88.18** | 84.39 | **73.49** | **90.71** | **82.59** | **79.13** | **94.84** | 72.12 |

French, Chinese), simulating real-world deployment scenarios; results are summarized in Table 4. Finally, we evaluate $D^3$ on Geometry3K to test multimodal self-evaluation capability, highlighting its applicability beyond text-only reasoning tasks. We analyze these results in detail below.

**Generalization Ability:** Table 1 reports the performance of our method across vertical-domain reasoning tasks of varying difficulty. For mathematical reasoning, $D^3$-Reasoning Score consistently achieves the highest improvements (+8.5%, +2.5%, +10.8%) across both the relatively easier GSM8K and the more challenging college-level TheoremQA benchmarks. For commonsense reasoning, $D^3$-Reasoning Score also demonstrates the strongest scoring capability across different model scales.

For multi-task scenarios that require both world knowledge and complex problem-solving skills, we report the performance of our approach on MMLU and MMLU-Pro in Table 2. These two massive multi-task benchmarks, each of which covers over 50 diverse downstream tasks, and we report the aggregated overall performance across all tasks. $D^3$-Reasoning Score shows strong generalization

Table 3: GSM8K results on larger models (Higher AUROC and AUPR indicate better performance; Lower FPR@95 indicates better performance).

| Dataset | Method | R1-Qwen-32B | | | QwQ-32B | | | R1-LLaMA-70B | | |
|---|---|---|---|---|---|---|---|---|---|---|
| | | AUROC ↑ | AUPR ↑ | FPR@95 ↓ | AUROC ↑ | AUPR ↑ | FPR@95 ↓ | AUROC ↑ | AUPR ↑ | FPR@95 ↓ |
| GSM8K | Random | 46.97 | 94.05 | 93.59 | 51.79 | 96.55 | 93.88 | 48.87 | 94.32 | 97.22 |
| | MaxProb | 54.66 | 94.88 | 94.81 | 66.18 | 97.07 | 72.34 | 60.09 | 96.14 | 85.71 |
| | PPL | 55.07 | 94.94 | 93.51 | 66.29 | 97.12 | 72.34 | 60.02 | 96.13 | 85.71 |
| | Entropy | 54.86 | 94.88 | 93.51 | 66.52 | 97.32 | 70.21 | 60.57 | 96.24 | 85.71 |
| | CoE-R | 62.51 | 96.62 | 96.10 | 66.93 | 97.64 | 85.11 | 61.55 | 96.33 | 88.57 |
| | CoE-C | 62.04 | 96.49 | 96.10 | 67.38 | 97.69 | 87.23 | 35.89 | 92.45 | 98.57 |
| | $D^3$ | **64.33** | **96.75** | 96.10 | **70.11** | **97.79** | **70.21** | 61.32 | 96.23 | **85.71** |

Table 4: Results on MGSM dataset across four languages using DeepSeek-R1-Distill-Qwen-7B (Arrows indicate direction of improvement).

| Model | Method | English (en) | | | German (de) | | | French (fr) | | | Chinese (zh) | | |
|---|---|---|---|---|---|---|---|---|---|---|---|---|---|
| | | AUROC ↑ | AUPR ↑ | FPR@95 ↓ | AUROC ↑ | AUPR ↑ | FPR@95 ↓ | AUROC ↑ | AUPR ↑ | FPR@95 ↓ | AUROC ↑ | AUPR ↑ | FPR@95 ↓ |
| DeepSeek-R1-Distill-Qwen-7B | Random | 54.12 | 91.52 | 100.00 | 45.08 | 74.66 | 96.83 | 52.67 | 78.79 | 95.12 | 49.42 | 68.94 | 95.89 |
| | MaxProb | 70.95 | 95.00 | 72.00 | 69.15 | 86.51 | 80.95 | 70.63 | 82.19 | 73.97 | 62.76 | 89.03 | 95.12 |
| | PPL | 70.92 | 94.98 | 72.00 | 69.36 | 86.62 | 80.95 | 70.91 | 82.86 | 73.97 | 63.25 | 89.01 | 92.68 |
| | Entropy | 73.24 | 95.40 | 68.00 | 69.15 | 87.04 | 80.95 | 70.27 | 82.04 | 72.60 | 65.14 | 90.13 | 92.68 |
| | CoE-R | 67.75 | 94.36 | 72.00 | 60.56 | 79.02 | 80.95 | 55.17 | 70.96 | 94.52 | 64.45 | 89.11 | 92.68 |
| | CoE-C | 66.58 | 94.04 | 72.00 | 58.18 | 78.08 | 85.71 | 54.15 | 70.21 | 98.63 | 63.43 | 88.61 | 95.12 |
| | $D^3$ | **74.03** | **95.27** | **64.00** | **69.44** | 86.91 | **79.37** | **71.32** | 82.36 | **72.60** | **71.04** | **91.28** | **92.68** |

ability (+3.3%, +5.7%, +2.3%) across all model scales on these challenging evaluations. Furthermore, in Table 2 we observe a consistent trend in the AUPR scores—emphasizing the precision of positive (correct) predictions—across different task difficulties and model sizes. As the difficulty increases from GSM8K to MMLU-Pro and model size varies, the AUPR scores of $D^3$-Reasoning Score behave in a statistically consistent manner: Higher AUPR scores (from 33.39 to 48.57 on MMLU-Pro and from 76.47 to 97.84 on MMLU) are observed as the task difficulty decreases or the model scale increases. This trend suggests that our method maintains stable and reliable behavior in diverse and realistic reasoning scenarios.

Beyond the results directly demonstrating the generality, the experiment in Figure 4 (Right part) conducts a more subtle comparison by analyzing the ROC curves against selected baselines. Rather than merely observing overall performance, this deeper analysis reveals that $D^3$-Reasoning Score consistently achieves lower FPR at the same TPR, indicating that it can maintain the same level of correct reasoning identification and even more effectively rejecting faulty outputs—a distinctive advantage not exhibited by other methods.

**Scalability:** In Table 3, we further report the performance of our approach on three stronger heterogeneous LRMs with model sizes exceeding 30B parameters. These large-scale models are widely adopted in high-performance and reliability-critical real-world scenarios. $D^3$-Reasoning Score consistently achieves the best or second-best scoring performance across all three models, demonstrating its strong capability even at larger model scales and under increasingly complex reasoning.

**Multi-lingual Applicability:** To evaluate the cross-lingual generalization of $D^3$-Reasoning Score, we conduct experiments on multilingual settings using MGSM subsets. As shown in Table 4, $D^3$-Reasoning Score consistently achieves the best performance across four popular languages: English, German, French, and Chinese. Notably, for certain languages such as Chinese, $D^3$-Reasoning Score even exhibits greater improvements than those observed in English. These results suggest that $D^3$ is capable of capturing semantically coherent reasoning dynamics across different linguistic structures.

**Multimodal Applicability.** We further validate the $D^3$ Reasoning Score in multimodal scenarios, where reasoning trajectories are more complex and incorporate signals from multiple modalities. Specifically, we train a multimodal reasoning model based on Qwen2.5-VL with the Geometry3K dataset under the Easy-R1 framework Yaowei Zheng (2025). Geometry3K requires integrating textual and visual information for spatial reasoning, thus serving as a challenging testbed for multimodal self-evaluation.

Table 5: Self-evaluation on Geometry3K.

| Method | AUROC ↑ | AUPR ↑ | FPR@95 ↓ |
|---|---|---|---|
| Random | 51.35 | 44.63 | 95.78 |
| MaxProb | 64.21 | 53.82 | 85.61 |
| PPL | 61.78 | 51.95 | 86.11 |
| Entropy | 66.21 | 55.61 | 87.24 |
| CoE-R | 69.13 | 59.60 | 82.80 |
| CoE-C | 57.42 | 46.73 | 90.50 |
| $D^3$ | **69.41** | **60.00** | **82.50** |

Table 6: Ablation study showing that our choice best captures reasoning trajectory across 4 models.

| Strategy | R1-Qwen-1.5B | | | R1-Qwen-7B | | | R1-LLaMA-8B | | | R1-Qwen-14B | | |
|---|---|---|---|---|---|---|---|---|---|---|---|---|
| | AUROC ↑ | AUPR ↑ | FPR@95 ↓ | AUROC ↑ | AUPR ↑ | FPR@95 ↓ | AUROC ↑ | AUPR ↑ | FPR@95 ↓ | AUROC ↑ | AUPR ↑ | FPR@95 ↓ |
| Mean Pooling ✔ | **77.28** | **91.42** | **72.20** | **76.32** | **94.73** | **67.27** | **68.12** | **89.37** | **80.31** | **61.64** | **94.04** | **76.19** |
| Last CoT token | 71.54 | 90.12 | 76.77 | 71.94 | 93.61 | 73.33 | 63.44 | 88.14 | 86.54 | 59.27 | 92.58 | 81.28 |
| Mean of all output tokens | 70.53 | 88.56 | 79.83 | 70.90 | 92.96 | 75.76 | 61.43 | 86.57 | 85.42 | 55.64 | 91.79 | 79.58 |

As shown in Table 7, $D^3$ achieves the best overall performance among all baselines, with consistent improvements across AUROC, AUPR, and FPR@95. These results highlight the robustness of $D^3$ in multimodal reasoning tasks, demonstrating that trajectory dynamics and difficulty calibration remain effective even when reasoning involves heterogeneous modalities. This confirms the general applicability of $D^3$ beyond text-only settings.

### 3.3 ABLATION STUDY

We further investigate how alternative design choices in $D^3$-Reasoning Score affect its performance. We focus on two key components: (i) the aggregation strategy for reasoning-relevant tokens, and (ii) the construction of $D^3$-Reasoning Score components. These components are critical because they directly determine how well the reasoning trajectory is captured and how accurately $D^3$-Reasoning Score reflects reasoning quality.

**Reasoning Token Aggregation Strategies:** Here, we investigate three alternative strategies to the mean pooling (i.e., selecting and aggregating the hidden representations over the reasoning tokens) introduced in Section 2.2 for constructing the $D^3$-Reasoning Score in Eq. (4). The first alternative way computes the mean hidden state across all generated reasoning tokens at each layer, using this mean as the semantic center of the model's ongoing reasoning process. The second alternative way directly uses the hidden state of the last generated CoT token, assuming it encapsulates the accumulated reasoning information up to that point. The third alternative way computes the mean hidden state over all output tokens generated from both reasoning phase and the final answer conclusion phase. Comparing these strategies aims to understand how different token aggregation schemes affect the fidelity of the modeled reasoning trajectory and ultimately the effectiveness of $D^3$-Reasoning Score in capturing reasoning soundness. Our ablation results in Table 6 show that mean pooling over reasoning tokens is the most effective aggregation strategy, capturing the core reasoning trajectory more reliably than the alternatives do.

$D^3$**-Reasoning Score Components.** We conduct an ablation study to analyze the contribution of each component in the $D^3$ formulation, namely $D_1$, $D_2$, and $D_3$ as defined in Section 2.2. Specifically, we evaluate how different combinations of these components influence the overall self-evaluation capability. Results on R1-7B are summarized in Table 7. The findings show that incorporating both $D_1$ and $D_2$ yields substantial gains over using either component alone, confirming the importance of modeling trajectory dynamics

Table 7: Ablation study of $D^3$ Reasoning Score Components

| $D_1$ | $D_2$ | $D_3$ | AUROC ↑ | AUPR ↑ | FPR@95 ↓ |
|---|---|---|---|---|---|
| ✓ | | | 74.35 | 92.60 | 72.12 |
| | ✓ | | 74.07 | 92.58 | 73.33 |
| ✓ | ✓ | | 75.23 | 93.60 | 72.73 |
| ✓ | | ✓ | 69.71 | 93.00 | 78.18 |
| | ✓ | ✓ | 70.11 | 93.07 | 78.18 |
| ✓ | ✓ | ✓ | **76.32** | **94.73** | **67.27** |

through both effective progress and adaptive adjustments. Moreover, adding the entropy term $D_3$ as a difficulty-aware calibrator further improves performance, producing the best overall self-evaluation results. Overall, the ablation demonstrates that all three components are indispensable for optimal performance: $D_1$ and $D_2$ jointly characterize trajectory dynamics, while $D_3$ provides necessary difficulty calibration. Together, they form a coherent and effective framework for white-box self-evaluation.

## 4 CONCLUSION

We introduced the $D^3$-Reasoning Score, a white-box self-evaluation metric that leverages hidden-state trajectory dynamics and difficulty calibration to assess the reliability of LRM outputs. By jointly modeling Distance, Dynamism, and Difficulty, $D^3$ provides a unified and interpretable measure of reasoning fidelity. Extensive experiments across diverse models, domains, and modalities demonstrate that $D^3$ consistently outperforms existing baselines in AUROC, AUPR, and FPR@95. These results establish $D^3$ as a principled and practical tool for systematic self-evaluation and diagnosis of reasoning trajectories, paving the way toward more reliable and trustworthy LRMs.

## REPRODUCIBILITY STATEMENT

To facilitate reproducibility, we provide the code in the supplementary material. The code repository includes detailed step-by-step instructions, allowing researchers to replicate our experiments with minimal effort. In addition, the model checkpoint obtained in our experiments will be made publicly available. Together, these resources ensure that our experimental results can be independently verified and extended by the research community.

## USE OF LARGE LANGUAGE MODELS

In accordance with the policy on the use of LLMs, we disclose that large language models were employed solely as auxiliary tools during the preparation of this work. Specifically, LLMs were used to (i) polish the writing for grammar and clarity, and (ii) assist in developing basic visualization scripts for presenting experimental results. Importantly, LLMs were not involved in the research ideation, experimental design, or analysis of findings. All scientific contributions and methodological innovations are entirely the work of the authors.

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

# A  APPENDIX

## A.1  LIMITATION

Although we have evaluated $D^3$-Reasoning Score on a wide range of state-of-the-art LRMs, the availability of open-sourced models remains limited, particularly for extremely large-scale LRMs. We hope that more high-quality open-sourced LRMs will become available in the future to facilitate broader exploration and validation of $D^3$-Reasoning Score at even larger model scales.

## A.2  RELATED WORK

**Large Reasoning Models (LRMs):** Comparing with LLMs, LRMs have demonstrated that extending the reasoning process can substantially enhance model inference performance Cui et al. (2025); Yuan et al. (2025). For example, OpenAI-o1 OpenAI (2024), Qwen-QwQ Team (2025b) and DeepSeek-R1 DeepSeek-AI (2025) explicitly perform CoT reasoning by constructing an intermediate reasoning trajectory trying to justify the final answer Wei et al. (2023). LRMs show to be effective in domains requiring complex logical deduction, such as mathematics and programming Chen et al. (2023); Cobbe et al. (2021); Hendrycks et al. (2021); Shi et al. (2022); Wang et al. (2024); Talmor et al. (2019); Wang et al. (2025a); Zhang et al. (2023); Yue et al. (2025); Zhao et al. (2025). Unlike conventional LLMs, which produce an answer in a single forward pass guided by shallow heuristics, LRMs follow an explicit "think-then-answer" routine DeepSeek-AI (2025); OpenAI (2024); Team (2025b;a); Muennighoff et al. (2025). They first enclose a CoT segment between special tokens (e.g., <think>...</think>) and repeatedly update internal representations while

attending to these intermediate tokens; They generate the final answer only after completing the step-by-step deliberation process.

Current efforts to evaluate the overall output quality of an LRM can be broadly categorized into external and internal assessment methods, which are separately reviewed as follows.

**External Assessment:** This type of assessment methods relies on supervision signals external to the model, such as linear probing techniques (Wang & Xu, 2025), additional annotations or knowledge bases (Li et al., 2023; Nguyen et al., 2024), and proxy models (He et al., 2025). For example, Kadavath et al. (2022) leverages the instruction-following capabilities of LLMs, enhanced by RLHF (Ouyang et al., 2022), to generate confidence scores through carefully crafted prompts. This paradigm encompasses a variety of general methodologies that directly elicit confidence estimates from LLMs via multi-stage pipelines, as broadly evaluated in (Lin et al., 2022; Tian et al., 2023). Manakul et al. (2023) further investigates this approach by prompting GPT-3 to generate biographical passages from the WikiBio dataset and manually annotating the factuality of the generated content. Similarly, the PSA pipeline (Xiong et al., 2024) estimates confidence by perturbing prompts (Gao et al., 2024) to produce multiple outputs and subsequently measuring their consistency. However, these methods typically require access to ground-truth labels, external resources, or auxiliary models, limiting their generalizability in real-world scenarios where such information may be unavailable or impractical to obtain. This limitation motivates the exploration of a label-free, model-internal evaluation strategies.

**Internal Assessment:** This type of assessment methods leverages the internal states obtained from the model itself, such as hidden representations (Wang et al., 2025b) or uncertainty-based signals (Shih et al., 2023; Huang et al., 2025; Si et al., 2023), to infer the reliability of its outputs without relying on external supervision. For example, Plaut et al. (2025) uses softmax probabilities to predict the correctness of multiple-choice questions, while Hendrycks et al. (2022) explores temperature scaling techniques for output probabilities. Kuhn et al. (2023) and Farquhar et al. (2024) investigate the concept of semantic entropy, aiming to capture uncertainty beyond surface-level probabilities. Furthermore, Duan et al. (2024) introduces a shift toward utilizing semantic information in entropy-based reliability estimation.

### A.3 MULTIMODAL VALIDATION ON GEOMETRY3K

We validate the $D^3$-Reasoning Score in a multimodal setting using the Geometry3K benchmark, which contains 3,002 geometry problems pairing textual statements with diagrams and formal annotations for symbolic reasoning. *Geometry3K* is introduced by Inter-GPS and targets integrated language–vision reasoning in Euclidean geometry. Lu et al. (2021) We employ **Qwen2.5-VL** (Instruct, 7B unless noted) as the vision–language backbone, following its official release notes and model repositories, which describe the series' multimodal encoder–decoder integration and available open-weight checkpoints (3B/7B/72B). Training and inference are orchestrated with **Easy-R1**, a clean fork of veRL designed for efficient, scalable, multi-modality RL/SFT workflows (with vLLM SPMD support).

For evaluation, each sample consists of the problem text, the diagram image, and multiple-choice options (A–D). We prompt the model to reason step by step and produce a single final option letter. During decoding, we log token-level hidden states over the reasoning segment and mean-pool per transformer layer to obtain $\bar{q}^i$, enabling computation of $D_1$, $D_2$, and the entropy-based difficulty term $D_3$. We then assess how well $D^3$ separates correct from incorrect predictions using AUROC/AUPR/FPR@95 on Geometry3K, demonstrating that the score remains informative when visual signals are integrated into the reasoning trajectory.

### A.4 VERBAL SELF-EVALUATION BASELINE

We additionally implemented a *verbal* self-evaluation baseline, where the model is prompted to output a confidence score in $[0, 1]$ after producing its answer. This approach is attractive because it is label-free and training-free; however, in our experiments it showed limited discriminative power for reasoning quality. Concretely, scores clustered in a narrow band ($\approx$0.95–1.0) for both correct and incorrect trajectories, yielding poor separation under AUROC/AUPR and high FPR@95. We hypothesize two causes: (i) verbalized confidence is systematically over-optimistic relative to ground

truth, and (ii) a single prompt format induces prompt-level bias that is largely decoupled from the underlying reasoning trajectory. As a result, this baseline was not included in every comparisons; we report it here for completeness and future replication.

---

**Self-Evaluation Prompt**

**Instruction:** Solve this math problem. Give the reasoning steps before giving the final answer on the last line by itself in the format of "Answer:". Do not add anything other than the integer answer after "Answer:".

**Then:** Provide a self-evaluation score between 0 and 1 of how confident you are in the correctness of your answer. Output the score in the format: "Score: <float between 0 and 1>"

**Question:**
`{input_data}`

---

For transparency, we release the exact prompt and evaluation script; researchers can reproduce the baseline and compare verbal scores with representation-based metrics like $D^3$ under identical settings.

## A.5 BENCHMARK DATASET DETAILS

We evaluate the generality and effectiveness of our proposed $D^3$-Reasoning Score across seven widely adopted reasoning benchmarks.

**GSM8K** Cobbe et al. (2021): a dataset designed for math word problem-solving tasks, involving arithmetic calculations and logical reasoning. Each question expects an integer as the final answer, accompanied by explicit reasoning steps. We follow standard evaluation metrics and report accuracy.

**CommonsenseQA** Talmor et al. (2019): a multiple-choice dataset evaluating commonsense reasoning. Each question provides five candidate answers (A–E), requiring the model to choose the correct option. Answers are evaluated by accuracy, emphasizing both factual knowledge and reasoning capability.

**TheoremQA** Chen et al. (2023): a dataset used to evaluate mathematical theorem reasoning and deduction abilities. It includes questions covering diverse mathematical topics, with answers in various formats such as numerical results, Boolean values, or lists of numbers. Performance is measured via accuracy.

**MMLU** Hendrycks et al. (2021): a dataset assessing models across diverse knowledge domains. It comprises multiple-choice questions from fields such as mathematics, sciences, and humanities, making it ideal for evaluating model versatility and general knowledge retrieval. Answers are evaluated using accuracy.

**MMLU-Pro** Wang et al. (2024): An extension of MMLU, MMLU-Pro introduces a more challenging set of questions to test robust multitask reasoning capabilities of LLMs. Similar to MMLU, questions span multiple disciplines and use accuracy as the evaluation metric.

**MGSM** Shi et al. (2022) MGSM is a multilingual benchmark focusing on math word problems. It evaluates arithmetic reasoning capabilities across multiple languages, including English, German, French, and Chinese. We evaluate performance using accuracy on numeric answers, demonstrating multilingual applicability.

**Geometry3K** Lu et al. (2021) Geometry3K is a large-scale geometry reasoning benchmark consisting of 3,002 geometry problems with dense annotation in formal language. Each problem combines a textual description and a geometric diagram, requiring models to perform integrated visual-textual reasoning over geometry constructions. In our experiments, we evaluate performance on Geometry3K by measuring accuracy of the extracted numeric or categorical answers, demonstrating our method's applicability to multimodal geometry reasoning tasks.

Detailed dataset statistics, task specifications, prompting templates, and illustrative examples for these benchmarks are provided in Table 8 and Table 9.

Table 8: Details of the downstream reasoning datasets (vertical-domain)

| Datasets (Train/Test) | GSM8K (7473/1319) | CommonsenseQA (9741/1140) | TheoremQA (-/800) |
|---|---|---|---|
| Venue | Cobbe et al. (2021) | Talmor et al. (2019) | Chen et al. (2023) |
| Task | Math word problem solving | Commonsense reasoning | Mathematical theorem reasoning |
| Metric | Accuracy (f) | Accuracy (f) | Accuracy (f) |
| Answer Format | Number | Option | Option / Boolean / Number / List of Numbers |
| Prompt Template | Solve this math problem. Provide reasoning steps before giving the final answer in the format "Answer:". Only output the integer after "Answer:".

**Question:** {input_data} | Answer the multiple choice commonsense question. Final line format: "Answer: $LETTER" (A/B/C/D/E). Provide step-by-step reasoning.

**Question:** {input_data} | Complete the math problem considering the provided answer type. Different formats are required for boolean, integer, list of numbers, or option.

**Question:** {input_data}
**Answer Type:** {answer_type} |
| Example | **Q**: Natalia sold clips to 48 friends in April, and half as many in May. How many in total?
**A**: 72 | **Q**: The sanctions against the school were a punishing blow. They seemed to what the efforts the school made to change? Options: A. ignore B. enforce C. authoritarian D. yell at E. avoid
**A**: A | **Q**: How many ways are there to divide a set of 8 elements into 5 non-empty ordered subsets?
**A**: 11760 |

Table 9: Details of the multi-domain and multilingual evaluation datasets

| Datasets (Train/Test) | MMLU (99842/14042) | MMLU-Pro (-/12032) | MGSM (-/250 per language) |
|---|---|---|---|
| Venue | Hendrycks et al. (2021) | Wang et al. (2024) | Shi et al. (2022) |
| Task | Multitask multiple-choice QA | Robust multitask multiple-choice QA | Multilingual math word problem solving |
| Metric | Accuracy (f) | Accuracy (f) | Accuracy (f) |
| Answer Format | Option | Option | Number |
| Prompt Template | Answer the multiple choice question. Final line format: "Answer: $LETTER".

**Question:** {input_data} | Same as MMLU.

**Question:** {input_data} | Solve the math problem. Output final answer after "Answer:".

**Question:** {input_data} |
| Example | **Q**: Let $p = (1, 2, 5, 4)(2, 3)$ in $S_5$. Find the index of $\langle p \rangle$ in $S_5$. Options: A. 8   B. 2   C. 24   D. 120
**A**: C | **Q**: Find the characteristic of the ring $2\mathbb{Z}$. Options: A. 0   B. 30   C. 3   D. 10   E. 12   F. 50   G. 2   H. 100   I. 20   J. 5
**A**: A | **Q**: A robe takes 2 bolts of blue fiber and half that much white fiber. How many bolts in total?
**A**: 3 |

## A.6 MODEL DETAILS

Table 10 summarizes the model configurations used in our experiments. We cover a range of model scales and architectures to validate the robustness and generality of $D^3$-Reasoning Score across diverse LLM backbones. The models differ in their base architectures, number of layers, hidden dimensions, and reserved special tokens (`<think>`, `</think>`) for explicit reasoning phase delineation during inference.

## A.7 DETAILED MATHEMATICAL DERIVATION OF EVALUATION METRICS

In this appendix, we provide the formal definitions and detailed derivations of the evaluation metrics used in this paper, namely AUROC, AUPR, and FPR@95, for assessing the alignment between $D^3$-Reasoning Score and the correctness of answers.

### A.7.1 PROBLEM SETUP AND NOTATIONS

We assume a dataset consisting of $B$ samples $\{(Q_i, \mathcal{E}_i, A_i^{\text{GT}})\}_{i=1}^B$, where:

- $Q_i$ denotes the $i$-th query.

- $\mathcal{E}_i$ denotes the $D^3$-Reasoning Score computed by our method for sample $i$.

- $A_i^{\text{GT}} \in \{0, 1\}$ is the ground-truth correctness label (1 for correct, 0 for incorrect).

Table 10: Model configuration details used in our experiments.

| Model Name | Base Model | Number of Layers | Hidden Size | Special Token IDs |
|---|---|---|---|---|
| DeepSeek-R1-Distill-Qwen-1.5B | Qwen2.5-Math-1.5B | 28 | 1536 | `<think>`: 151648
`</think>`: 151649 |
| DeepSeek-R1-Distill-Qwen-7B | Qwen2.5-Math-7B | 28 | 3584 | `<think>`: 151648
`</think>`: 151649 |
| DeepSeek-R1-Distill-Llama-8B | Llama-3.1-8B | 32 | 4096 | `<think>`: 128013
`</think>`: 128014 |
| DeepSeek-R1-Distill-Qwen-14B | Qwen2.5-14B | 48 | 5120 | `<think>`: 151648
`</think>`: 151649 |
| DeepSeek-R1-Distill-Qwen-32B | Qwen2.5-32B | 64 | 5120 | `<think>`: 151648
`</think>`: 151649 |
| DeepSeek-R1-Distill-Llama-70B | Llama-3.3-70B-Instruct | 80 | 8192 | `<think>`: 128013
`</think>`: 128014 |
| QwQ-32B | Qwen2.5-32B | 64 | 5120 | `<think>`: 151667
`</think>`: 151668 |

We further define:

- $\mathcal{P} = \{i \mid A_i^{\text{GT}} = 1\}$ : set of indices for positive (correct) samples.
- $\mathcal{N} = \{j \mid A_j^{\text{GT}} = 0\}$ : set of indices for negative (incorrect) samples.

### A.7.2 AUROC (AREA UNDER RECEIVER OPERATING CHARACTERISTIC)

AUROC measures the probability that a randomly chosen positive sample is ranked higher than a randomly chosen negative sample based on the $D^3$-Reasoning Score. It is formally computed as:

$$\text{AUROC} = \frac{1}{|\mathcal{P}||\mathcal{N}|} \sum_{i \in \mathcal{P}} \sum_{j \in \mathcal{N}} \Bbbk(\mathcal{E}_i > \mathcal{E}_j), \tag{5}$$

where $\Bbbk(\cdot)$ is the indicator function:

$$\Bbbk(\mathcal{E}_i > \mathcal{E}_j) = \begin{cases} 1, & \text{if } \mathcal{E}_i > \mathcal{E}_j, \\ 0, & \text{otherwise.} \end{cases} \tag{6}$$

A higher AUROC (closer to 1.0) indicates better global ranking capability.

### A.7.3 AUPR (AREA UNDER PRECISION-RECALL CURVE)

AUPR quantifies the precision-recall trade-off, especially important under class imbalance. At a threshold $t$, precision and recall are defined as:

$$\text{Precision}(t) = \frac{TP(t)}{TP(t) + FP(t)}, \tag{7}$$

$$\text{Recall}(t) = \frac{TP(t)}{TP(t) + FN(t)}, \tag{8}$$

where $TP(t)$, $FP(t)$, and $FN(t)$ denote true positives, false positives, and false negatives at threshold $t$, respectively.

AUPR is computed as the area under the precision-recall curve:

$$\text{AUPR} = \int_0^1 \text{Precision}(\text{Recall}) \, d(\text{Recall}), \tag{9}$$

which is typically approximated using numerical integration methods (e.g., trapezoidal rule).

### A.7.4 FPR@95 (FALSE POSITIVE RATE AT 95% RECALL)

FPR@95 measures the false positive rate at the threshold where the recall reaches exactly $95\%$, evaluating robustness under stringent correctness requirements.

We first find the decision threshold $t^*$ such that:

$$t^* = \arg\min_t |\text{Recall}(t) - 0.95|. \tag{10}$$

Then, the FPR@95 is computed as:

$$\text{FPR@95} = \frac{FP(t^*)}{|\mathcal{N}|}. \tag{11}$$

Lower FPR@95 indicates that the $D^3$-Reasoning Score reliably identifies correct reasoning trajectories with minimal false positives even at high-recall thresholds.

### A.8 INTERESTING DISCUSSION: PHYSICAL PROJECTIONS OF $D^3$

In developing $D^3$, we observed that its formulation resonates with certain dynamical concepts in physics. While this connection should not be regarded as a strict proof or formal analysis, it offers an interesting perspective that may inspire further thinking. Importantly, this analogy is entirely independent of the core design and validity of $D^3$.

#### A.8.1 SETUP AND NOTATION

Let $\bar{q}^i \in \mathbb{R}^d$ denote the layer-averaged hidden state at the $i$-th reasoning step ($i = 0, \ldots, K$). Define discrete first- and second-order increments

$$\Delta \bar{q}^i = \bar{q}^i - \bar{q}^{i-1}, \tag{12}$$

$$\Delta^2 \bar{q}^i = \left(\bar{q}^i - \bar{q}^{i-1}\right) - \left(\bar{q}^{i-1} - \bar{q}^{i-2}\right) = \Delta \bar{q}^i - \Delta \bar{q}^{i-1}. \tag{13}$$

The three components of $D^3$ are

$$D_1 = \sum_{i=1}^{K} \left\|\Delta \bar{q}^i\right\|, \qquad D_2 = \sum_{i=2}^{K} \left\|\Delta^2 \bar{q}^i\right\|, \qquad D_3 = \frac{1}{K} \sum_{k=1}^{K} \left[-\log p_k\right], \tag{14}$$

where $p_k$ is the model probability assigned to the ground-truth token at step $k$. The overall score is

$$D^3 = D_1 + D_2 - D_3. \tag{15}$$

#### A.8.2 PROJECTION I: KINEMATICS AND PATH GEOMETRY

**Distance as path length.** $D_1$ aggregates $\|\Delta \bar{q}^i\|$, a discrete arc-length of the trajectory $\{\bar{q}^i\}_{i=0}^{K}$ in representation space. In kinematics, arc-length is induced by instantaneous speed; here, $\|\Delta \bar{q}^i\|$ is the magnitude of a one-step representational shift. Large $D_1$ indicates substantial *effective progress* along the reasoning path.

**Dynamism as curvature/acceleration.** $D_2$ aggregates $\|\Delta^2 \bar{q}^i\|$, a discrete analogue of acceleration or curvature-driven change of direction. In trajectory terms, $D_2$ rewards *adaptive adjustments*: variable step sizes and direction shifts that reflect exploratory refinements rather than rigid, straight-line drift. This mirrors the intuition that successful multi-step reasoning often requires turning, branching, or correcting course.

**Combined trajectory dynamics.** $D_1 + D_2$ thus measures both *how far* the representation travels (progress) and *how adaptively* it moves (dynamism). Empirically, easier queries tend to admit higher $D_1 + D_2$ (more decisive, adaptive exploration), whereas harder queries often exhibit flatter trajectories.

### A.8.3 PROJECTION II: ENERGETICS AND BARRIERS

**Difficulty as uncertainty/potential barrier.** $D_3$ is the predictive entropy averaged along the trajectory. In a thermodynamic reading, entropy here serves as an *uncertainty potential*: higher $D_3$ marks intrinsically harder queries, a larger "barrier" to be overcome by trajectory dynamics.

**A free-energy–like form.** Writing $S_{\mathrm{dyn}} := D_1 + D_2$ and $U := D_3$, the score becomes

$$D^3 = S_{\mathrm{dyn}} - U, \tag{16}$$

which echoes a free-energy decomposition (useful work capacity = drive − barrier). While purely analogical, this perspective clarifies why difficulty calibration is essential: for fixed dynamics, increasing uncertainty should *decrease* the confidence of self-evaluation.

### A.8.4 PROJECTION III: ACTION–LIKE FUNCTIONALS (HEURISTIC)

In classical mechanics, the action is $\int (T - V)\, dt$. Discretely, one may view

$$\mathcal{S} \approx \sum_{i=1}^{K} \underbrace{\left\lVert \Delta \bar{q}^i \right\rVert}_{\text{progress}} + \sum_{i=2}^{K} \underbrace{\left\lVert \Delta^2 \bar{q}^i \right\rVert}_{\text{adaptation}} - \frac{1}{K} \underbrace{\sum_{k=1}^{K} \left[ -\log p_k \right]}_{\text{uncertainty}}, \tag{17}$$

which is precisely $D^3$. The first two terms encourage sufficient motion and corrective steering; the last term penalizes high uncertainty. This is a mnemonic design lens, not a physical derivation.

### A.8.5 INVARIANCES, SCALING, AND REGULARITY

- **Translation invariance:** $D_1$ and $D_2$ depend only on differences of $\bar{q}^i$, hence are invariant to global translations $\bar{q}^i \mapsto \bar{q}^i + c$.
- **Monotone calibration:** Increasing $D_3$ (harder queries) should reduce the score; the $-D_3$ term enforces this monotonicity.
- **Step reparameterization:** If reasoning emits more (or fewer) steps for the same underlying move, $D_1$ and $D_2$ balance path length versus curvature; aggregating both mitigates pathological resampling that would fool single-order metrics.

### A.8.6 WHAT INCREASES THE SCORE?

1. Larger *effective progress* ($D_1 \uparrow$) along semantically meaningful directions.
2. Appropriate *adaptive adjustments* ($D_2 \uparrow$) that reflect purposeful exploration and refinement.
3. Lower *uncertainty* ($D_3 \downarrow$) consistent with confident, well-supported reasoning.

High $D^3$ thus corresponds to a trajectory that travels far enough, turns when necessary, and remains certain—hallmarks of sound multi-step reasoning.

### A.8.7 A COMPACT MAPPING

| Component | Geometry/Kinematics | Physics Projection | Interpretation in $D^3$ |
|---|---|---|---|
| $D_1$ | Arc-length ($\sum \lVert \Delta \bar{q} \rVert$) | Speed / path extent | Effective progress |
| $D_2$ | Discrete curvature ($\sum \lVert \Delta^2 \bar{q} \rVert$) | Acceleration / turning | Adaptive exploration |
| $D_3$ | Predictive entropy | Potential barrier / uncertainty | Difficulty calibrator |

Table 11: Heuristic projection of $D^3$ onto geometric and physical notions.

### A.8.8 CAVEATS AND SCOPE

The projections above are analogies for intuition-building. They are *not* claims of physical law inside hidden-state space. Our conclusions and empirical gains do not hinge on this viewpoint; the $D^3$ score is defined operationally from model-internal signals and validated by its predictive alignment with correctness.

**Takeaway.** $D^3 = D_1 + D_2 - D_3$ can be read as *trajectory dynamics minus uncertainty barrier*. This lens cleanly explains why difficulty calibration is necessary and why combining first- and second-order dynamics is more informative than either alone—while keeping the interpretation faithful to the underlying geometry of hidden-state evolution.

## A.9 PROMPTS USED PER DATASET

Below we enumerate the exact prompts we used for each dataset during inference. For clarity, each prompt is shown inside a blue framed box.

---

**GSM8K Prompt**

Solve this math problem. Give the reasoning steps before giving the final answer on the last line by itself in the format of "Answer:".
Do not add anything other than the integer answer after "Answer:".
Question:
`{input_data}`

---

**mgsm Prompt**

Solve this math problem. Giving the final answer by itself in the format of "Answer:".
Do not add anything other than the integer answer after "Answer:".
Question:
`{input_data}`

---

**mgsm_fr Prompt**

Résolvez ce problème de mathématiques. Donnez la réponse finale sur la dernière ligne elle-même dans le format de "Réponse:".
N'ajoutez rien d'autre que la réponse entière après "Réponse:".
Question:
`{input_data}`

---

**mgsm_de Prompt**

Löse dieses Mathematikproblem. Gib die endgültige Antwort in der letzten Zeile alleine im Format "Antwort:" an.
Füge nichts anderes als die ganzzahlige Antwort nach "Antwort:" hinzu.
Frage:
`{input_data}`

---

**mmlu Prompt**

Answer the following multiple choice question. The last line of your response should be of the following format: "Answer: LETTER" (without quotes) where LETTER is one of A/B/C/D.
Question:
`{input_data}`

---

### CommonsenseQA Prompt

Answer the following multiple choice common-sense reasoning question. The last line of your response should be of the following format: "Answer: LETTER" (without quotes) where LETTER is one of A/B/C/D/E. Think step by step and output the reasoning process before answering.

`{input_data}`

### TheoremQA Prompt

Below is an instruction that describes a task, paired with an input that provides further context. Write a response that appropriately completes the request.

**Instruction:** Read a math problem. If Answer Type is [bool], respond True/False. If numeric, respond accordingly. If list, respond like [2,3,4]. If option, respond like (a)/(b)/(c)/(d). Use final sentence: "Therefore, the answer is . . . ".

Question: `{input_data}`

Answer_type: `{answer_type}`

Response:

