# OpenReview forum: "Reasoning Self-Evaluation via Trajectory Dynamics Modeling"
_ICLR.cc/2026/Conference — Submitted to ICLR 2026_

### Official Review · Reviewer_kDen · 2025-10-30

**Soundness:** 2
**Presentation:** 3
**Contribution:** 2
**Rating:** 2
**Confidence:** 4

**Summary:**

This paper proposes the D³-Reasoning Score, a metric designed to evaluate the reliability of reasoning in Large Reasoning Models (LRMs) by analyzing their internal hidden-state dynamics. D³ combines layer-wise representation changes (D₁, D₂) and predictive uncertainty of reasoning tokens (D₃) to enable white-box self-evaluation of reasoning quality.

**Strengths:**

Strengths

- This paper is well written and easy to follow.

- The motivation—to enable white-box, self-evaluative reasoning in large reasoning models—is well presented and addresses an important gap in current interpretability and reliability research. The authors convincingly argue for the need to go beyond external evaluation toward internal reasoning analysis.

- The paper provides empirical evidence that the proposed D³-Reasoning Score improves self-evaluation robustness and correlates with reasoning reliability across multiple benchmarks.

**Weaknesses:**

### Weaknesses

- **Questionable theoretical grounding of D³ components.**
   The layer-wise metrics (**D₁**, **D₂**) assume that transformer depth corresponds to reasoning progression, which is conceptually inconsistent since layers represent representational hierarchy rather than temporal reasoning steps. As a result, these measures likely capture architectural transformations or activation scaling rather than true reasoning dynamics.

- **Ambiguity in defining the predictive entropy term (D₃).**
   Best of my understanding, the definition of **D₃** relies on log-probabilities of “ground truth” tokens during the reasoning phase, but no such supervision exists within the `<think>` segment, especially for the test dataset which we cannot access to the ground truth. Also, averaging over the entire trajectory penalizes self-corrective reasoning patterns (e.g., “aha” moments) and conflates transient uncertainty with task difficulty.

- **Lack of normalization and inconsistent scaling across components.**
   The final score, \( D^3 = D_1 + D_2 - D_3 \), combines quantities with incompatible units—hidden-state L2 norms and log-probabilities—without normalization or weighting. This linear aggregation lacks theoretical justification and likely reflects empirical tuning rather than principled reasoning modeling.

- **Potential over-interpretation of empirical trends.**
   The observed correlation between D³ and dataset difficulty may result from incidental activation geometry rather than intrinsic reasoning reliability. The metric appears heuristic and architecture-dependent, raising doubts about its generalizability beyond the reported experiments.

- **Unbalanced and partially unreliable presentation of experimental results.**
   In several result tables, the authors only highlight cases where the proposed model ranks first or second, omitting markings for baselines that perform equally well or even better. Moreover, standard deviations are not reported, despite many results showing very similar scores across methods. This selective and incomplete reporting weakens the reliability and transparency of the experimental claims.

**Questions:**

See above section

---

> ### Author Response · Authors · 2025-11-25
> **Response to Reviewer kDen (1)**
>
> We sincerely thank you for your comprehensive review. We appreciate your thoughtful concerns and present detailed point-by-point responses below. We hope these clarifications will address your concerns and would be grateful if your position to this work could get better.
>
> > **Weakness 1: Questionable theoretical grounding of D³ components**
>
> #### Clarifying the intent of D₁ and D₂
>
> Thank you for highlighting the conceptual distinction between representational reasoning progression (transformer depth) and temporal reasoning progression. Your comment drives us to clarify the design motivation behind the **Distance (D₁)** and **Dynamism (D₂)** components.
>
> The central challenge in self-evaluating reasoning lies precisely in the lack of a metric that directly matches the latent notion of “reasoning quality.” Currently, it is extremely difficult to design a metric or theoretical mechanism that directly describes reasoning quality. If such an oracle metric existed, there would be no need to resort to internal representational changes—the problem would be trivial. Consequently, existing work aims to establish a linkage between observable metrics and reasoning quality. As you rightly pointed out, profiling reasoning progression is a promising direction for reflecting such a linkage.
>
> As a reasoning progression profiling method, our work stems from an intuitive yet empirically testable hypothesis: **the evolution of intermediate representations within the model serves as a proxy for reasoning progression in latent space and correlates with the quality of its thinking.** This motivates us to systematically investigate the relationship between representational dynamics and reasoning quality. Our findings demonstrate that this linkage not only exists but can be quantified in a way that **consistently reflects the quality of the reasoning process across tasks and models.**
>
> You raised a valid concern that our Trajectory Dynamics Modeling method might not be consistent with "temporal reasoning steps" in the conventional sense of reasoning progress. We indeed construct this linkage from a novel perspective—the representational reasoning progression within the model. Prior works that profile reasoning progress in a temporal sequence manner, such as MaxProb and PPL, often fail to establish a solid linkage to reasoning quality. This suggests that temporal reasoning steps alone may not be a sufficient metric for capturing "true reasoning dynamics." As demonstrated in the experiment below, you will see that using (token-wise) temporal reasoning does not work.
>
> As described in Section 2.1 of the paper, our approach considers layerwise hidden-state changes as reflecting the model’s **internal progression** during reasoning, even though this progression is not explicitly temporal. Each transformer layer applies an attention–feedforward transformation that continuously *refines* the intermediate representation by integrating broader context and accumulated semantic cues from earlier layers.
> This process can be interpreted as another form of **reasoning progression**, analogous to iterative optimization. Upon arriving at the deeper layers, the representation associated with each reasoning token has incorporated richer contextual and high-level abstractions. Thus, deeper layers often encode **more semantically convergent or stabilized representations**, which naturally serve as proxies for “more advanced” reasoning states.
>
> From the model’s perspective, the true progression lies in the **evolution of its hidden-state geometry**, not in wall-clock token emission. The real reasoning progression happens *within* each forward pass, along the depth dimension, where multi-level representational transformations accumulate. This is precisely the trajectory dynamics that D₁ and D₂ measure. They are therefore intentionally defined over **layerwise representational updates**, rather than external time.
> #### Analogy Table
>
> | Domain | Space | Progression Indicator | Dynamics |
> |--------|--------|------------------------|-----------|
> | Real World | Physical coordinates (x, y, z) | Temporal axis (seconds, minutes, …) | Velocity, acceleration |
> | **LRM Trajectory Dynamics** | Latent representation space | Layerwise representational refinement | D₁ (first-order), D₂ (second-order) |
>
> To further clarify our perspective, we provide the analogy above. Transformer layers do not correspond to temporal steps in a human sense; however, they *do* define a sequence of increasingly refined representations in latent space. This representational evolution forms a meaningful trajectory that can be analyzed in a manner analogous to dynamics in continuous systems.
>
> (continued on next page)

---

> ### Author Response · Authors · 2025-11-25
> **Response to Reviewer kDen (2)**
>
> (continued from previous page)
>
> #### Token-wise (Temporal) Dynamics as an Alternative
>
> To directly address your concern, we implemented the alternative paradigm you suggested—**token-wise (temporal) dynamics**, where representational reasoning progression is modeled along the *K* dimension instead of the *L* dimension.
>
> | Model | Method | GSM8K | | | CommonsenseQA | | | TheoremQA | | |
> |-------|--------|-------|------|---------|---------------|------|---------|-----------|------|---------|
> | | | AUROC ↑ | AUPR ↑ | FPR@95 ↓ | AUROC ↑ | AUPR ↑ | FPR@95 ↓ | AUROC ↑ | AUPR ↑ | FPR@95 ↓ |
> | R1-Qwen-1.5B | Random | 50.23 | 75.39 | 94.29 | 50.03 | 46.81 | 93.14 | 49.75 | 40.35 | 92.81 |
> | | +Temporal Dynamics | 51.84 | 73.92 | 96.17 | 48.36 | 48.15 | 94.73 | 51.29 | 39.18 | 94.26 |
> | | Verbal | 51.48 | 75.11 | 96.34 | 52.01 | 45.87 | 91.81 | 51.51 | 39.94 | 93.22 |
> | | MaxProb | 70.34 | 87.97 | 76.95 | 64.06 | 61.16 | 81.42 | 53.65 | 43.91 | 92.19 |
> | | PPL | 70.88 | 88.24 | 77.29 | 63.86 | 61.07 | 83.19 | 54.05 | 44.24 | 91.25 |
> | | Entropy | 71.86 | 88.54 | 75.93 | 64.26 | 61.32 | 82.30 | 54.33 | 44.52 | 92.19 |
> | | CoE-R | 74.17 | 89.99 | 75.93 | 60.03 | 56.37 | 85.18 | 53.33 | 44.18 | 93.44 |
> | | CoE-C | 73.32 | 89.37 | 75.25 | 58.90 | 55.44 | 85.18 | 52.42 | 43.34 | 90.31 |
> | | **D³** | **77.28** | **91.42** | **72.20** | **64.26** | **61.34** | **81.19** | **54.37** | **44.82** | **90.00** |
> | R1-LLaMA-8B | Random | 50.22 | 81.21 | 95.33 | 51.75 | 77.22 | 96.14 | 46.29 | 41.67 | 92.91 |
> | | +Temporal Dynamics | 48.93 | 82.64 | 96.88 | 53.12 | 75.49 | 94.87 | 47.81 | 40.29 | 91.53 |
> | | Verbal | 51.22 | 85.49 | 94.77 | 51.04 | 75.95 | 97.41 | 49.11 | 40.24 | 93.57 |
> | | MaxProb | 58.85 | 85.46 | 94.49 | 74.71 | 89.29 | 67.38 | 59.39 | 53.12 | 84.65 |
> | | PPL | 58.94 | 85.51 | 94.49 | 74.55 | 89.23 | 67.81 | 60.01 | 53.91 | 85.04 |
> | | Entropy | 60.25 | 85.89 | 94.88 | 74.68 | 89.28 | 65.24 | 60.27 | 54.06 | 84.65 |
> | | CoE-R | 63.01 | 86.57 | 86.22 | 52.00 | 77.92 | 93.56 | 50.56 | 46.02 | 96.46 |
> | | CoE-C | 63.86 | 86.50 | 86.61 | 73.90 | 88.92 | 69.96 | 58.01 | 54.10 | 93.70 |
> | | **D³** | **68.12** | **89.37** | **80.31** | **74.71** | **89.35** | **65.24** | **60.57** | **56.06** | **83.86** |
> | R1-Qwen-14B | Random | 49.46 | 93.23 | 95.24 | 49.13 | 84.30 | 95.03 | 50.93 | 56.79 | 94.93 |
> | | +Temporal Dynamics | 51.17 | 91.86 | 96.52 | 47.58 | 85.71 | 93.46 | 49.37 | 58.24 | 96.18 |
> | | Verbal | 50.44 | 91.91 | 96.14 | 54.19 | 85.87 | 94.85 | 51.83 | 56.54 | 94.72 |
> | | MaxProb | 53.98 | 93.06 | 85.71 | 78.90 | 94.87 | 65.22 | 51.94 | 57.71 | 89.40 |
> | | PPL | 54.51 | 93.23 | 84.52 | 79.24 | 94.99 | 64.60 | 52.39 | 58.07 | 89.40 |
> | | Entropy | 55.13 | 93.26 | 85.71 | 79.24 | 94.97 | 65.84 | 52.62 | 58.24 | 90.32 |
> | | CoE-R | 59.62 | 93.84 | 77.38 | 71.78 | 91.82 | 73.29 | 52.28 | 58.32 | 93.55 |
> | | CoE-C | 58.63 | 93.66 | 77.38 | 68.40 | 90.78 | 80.75 | 50.99 | 57.39 | 92.63 |
> | | **D³** | **61.64** | **94.04** | **76.19** | **79.27** | **95.00** | **61.49** | **52.65** | **58.77** | **88.02** |
>
> #### Results of the Token-wise Variant
> The results are decisive: **the token-wise dynamics variant performs poorly across all evaluation metrics**, often only slightly above the random baseline and consistently **worse than our layer-wise D³ formulation**. In particular, the “+Temporal Dynamics” rows:
>
> - show AUROC values close to 50 on several settings,
> - yield higher FPR@95 than random in many cases, and
> - do not approach the performance of our layer-wise D³ score.
>
> This confirms that simply following hidden states along the token sequence does **not** provide a reliable signal for self-evaluation.
>
> 1. **Token-level dynamics are dominated by local semantic variation rather than reasoning progress.**
>    Each token’s hidden state primarily captures the semantics of that token and its immediate context. The change from one token to the next often reflects changes in *lexical or phrasal content* rather than meaningful stages of reasoning.
>    For example, in a phrase like “red apple”, the transition from “red” to “apple” induces a large representational shift that is almost entirely semantic (“color” → “object”), not a change in *reasoning state*. In long CoT outputs, such semantic shifts between neighboring tokens are ubiquitous.
>    As a result, token-wise hidden state changes are highly instance-specific and sensitive to surface wording (e.g., “hmm”, “let me think”, self-corrections). These idiosyncratic fluctuations introduce substantial **semantic noise** that is weakly correlated with whether the overall reasoning is correct or not.
>
> (continued on next page)

---

> ### Author Response · Authors · 2025-11-25
> **Response to Reviewer kDen (3)**
>
> (continued from previous page)
>
>
> 2. **Lack of alignment and comparability across examples.**
>    Different queries produce reasoning traces with very different lengths and structures. A given token index (e.g., the 5th or 100th token) does **not** correspond to a comparable stage of reasoning across problems: in one example the 100th token may appear early in a long chain, while in another it is already near the conclusion.
>    Using $K$ as the evolution axis therefore yields trajectories of varying length with **no natural alignment** across samples. Any summary statistic over such trajectories mixes heterogeneous “stages” of reasoning and severely harms comparability and generalization.
>    In contrast, every input is processed by exactly the same sequence of layers, so layer $m$ always represents “after $m$ rounds of internal computation.” This provides a **consistent and comparable progression axis** across all examples, which is essential for a general-purpose self-evaluation metric like D³.
>
> Taken together, these observations indicate that **temporal token transitions do not reliably correspond to reasoning progression**. The model’s meaningful internal representational progression occurs *within each forward pass*, across layers, not across output tokens.
>
> #### Prior Work Supporting Internal Dynamics as Indicators of Reliability
>
> Our approach builds on a rapidly growing body of evidence that **internal model activations carry strong signals about reasoning reliability, correctness, and uncertainty**. Several recent studies support this view:
>
> - **Azaria & Mitchell (2023)** [1]
>   They demonstrate that hidden activations encode whether the model is telling the truth or lying. Probes trained on hidden states predict truthfulness far above chance, illustrating that internal representations contain correctness-related signals even when the model’s surface output may not.
>
> - **Burns et al. (2024)** [2]
>   These authors discover a linear “truthfulness direction” in hidden-state space that separates true from false statements. Using only internal activations, they are able to answer questions more reliably than the model’s zero-shot outputs. This indicates that deeper-layer activations encode factual reliability.
>
> - **Liu et al. (2023)** [3]
>   In their work on “cognitive dissonance,” they show that hidden-state probes outperform output probabilities in predicting correctness and provide better calibration. They conclude that much of the model’s latent knowledge is stored internally, not faithfully expressed by the output layer.
>
> - **Wang et al. (2025b)** [4]
>   This recent study analyzes hidden-state for self-evaluation, showing that internal embedding distributions correlate with whether the reasoning trajectory will yield a correct answer.
>
> Collectively, these works reinforce that transformer internal representations encode rich information about reasoning quality, truthfulness, and uncertainty. Our D₁/D₂ formulation is fully aligned with this evidence: it extracts *meaningful latent reliability signals* consistently observed in the literature.
>
>
> #### Key Insight
>
> Temporal reasoning steps and our Trajectory Dynamics Modeling represent two orthogonal approaches to profiling reasoning progression, both aiming to link dynamics to reasoning quality. Our results demonstrate that Trajectory Dynamics Modeling yields significantly better performance.
>
> We will further revise the manuscript to more clearly emphasize that **the meaningful notion of trajectory dynamics in LRMs lies in the model’s *internal* layerwise evolution**.
>
> We hope this resolves your concern and clarifies why the D³ formulation is grounded in a principled understanding of internal reasoning dynamics rather than temporal assumptions.
>
>
> ---
>
> #### References
>
> [1] **Azaria, A. & Mitchell, T.** *The internal state of an LLM knows when it’s lying.* EMNLP 2023.
> [2] **Burns, C. et al.** *Discovering latent knowledge in language models without supervision.* ICLR 2023.
> [3] **Liu, K. et al.** *Cognitive dissonance: Why LLM outputs disagree with internal representations of truthfulness.* EMNLP 2023.
> [4] **Wang, Y. et al.** *Latent Space Chain-of-Embedding Enables Output-Free LLM Self-Evaluation.* ICLR 2025.

---

> ### Author Response · Authors · 2025-11-25
> **Response to Reviewer kDen (4)**
>
> > **Weakness 2: Ambiguity in defining the predictive entropy term (D₃)**
>
> We appreciate the reviewer’s careful reading and agree that our wording around “ground‑truth token” in Section 2.2.2 was ambiguous. Below we clarify the definition and role of D₃ concisely.
>
>
> #### Clarified Definition of D₃
>
> In our design, the “Difficulty” term in the D³ score is instantiated via Shannon-entropy of the model’s token probability distribution along the reasoning trajectory. Specifically, as described in Section 2.2.2 (“Entropy-Based Query Difficulty Calibration”), we define:
>
> $$
> D_3 = \mathbb{E}_{1 \le k \le K}[-\log p_k]
> $$
>
> where $p_k$ denotes the probability of the selected k-th token in the reasoning trajectory.
> Here we clarify the derivation from the expectation form to the explicit average over the full sequence:
> $D_3 = \frac{1}{K} \sum_{k=1}^K -\log p_k
> $
>
> Intuitively, $D_3$ captures the **average uncertainty** of the distribution encountered at each step of the chain: higher values of $D_3$ correspond to greater uncertainty, hence 'harder' queries.
> We understand the ambiguity around the term *"correct token"/"ground-truth token"*. In fact, we meant the token that was *actually selected* in the reasoning trajectory (i.e., the model's own output token at k), rather than a teacher-forced target sequence. In other words, we compute the Shannon‐entropy value for each step in the reasoning chain and then take the **average over the entire sequence of length $K$**.
>
> (continued on next page)
>
> To further clarify, our implementation is publicly available (in the supplemental material and the anonymised repository). One may inspect the `compute_entropy` function in `score.py`. Here, `self.all_token_re` represents the list of token-probability trajectories of the sampled inference path, and `entropy(re, base=2)` computes the Shannon entropy of that trajectory. Thus, the "correct token" is in fact the token output by the model itself at each reasoning step, not a teacher-forced target. We hope this further clarification helps eliminate any ambiguity around the definition.
>
>
> #### On "aha" moments, transient uncertainty, and difficulty
>
> We now address the core of your concern: that averaging uncertainty over the trajectory might mistake **self‑correction (a common LRM behavior)** for **intrinsic difficulty**, and therefore decouple $D_3$ from genuine task hardness.
>
>  From a reliability perspective, a query that requires long, unstable exploration before converging is more fragile under perturbations than one that is confident throughout—even if both are eventually correct. It is therefore *desirable* that such a kind of trajectories receive a higher $D_3$, meaninig more difficult reasoning: they are exactly the borderline cases where the model is close to failure.
>
> **Not just "style": dataset‑level evidence.**
> If $D_3$ were driven mainly by a generic self‑correction style, independent of hardness, its distribution would not systematically track dataset difficulty. However, as shown in our new table below, the empirical distribution of $D_3$ shifts monotonically upward as benchmark complexity increases, while architecture, decoding, and prompting are fixed. This indicates that high average entropy is not just capturing a ubiquitous "exploratory habit" of the model, but aligns with tasks that are genuinely harder *for this LRM*.

---

> ### Author Response · Authors · 2025-11-25
> **Response to Reviewer kDen (5)**
>
> (continued from previous page)
>
>
> **Table: Entropy Distribution Across Datasets of Varying Difficulty**
>
> | Dataset  | # Samples | Median Entropy | 75th %ile | 90th %ile | 95th %ile | Range |
> |---------|-----------|----------------|-----------|-----------|-----------|-------|
> | **GSM8K** | 1,319 | 0.232 | 0.275 | 0.325 | 0.363 | [0.095, 0.921] |
> | **CommonsenseQA**   |1,140 | 1.544 | 1.760 | 1.976 | 2.084 | [0.357, 2.515] |
> | **MMLU** | 14,042  | 1.905 | 2.049 | 2.169 | 2.234 | [1.035, 2.454] |
>
> *All measurements use R1-Qwen-1.5B with identical prompting and decoding settings.*
>
> **Key observations:**
> 1. **Monotonic shift across difficulty levels**: As task complexity increases, the median entropy increases by **6.6×** (0.232 → 1.544) and **8.2×** (0.232 → 1.905), respectively.
> 2. **Consistent distributional separation**: The 75th and 90th percentiles show clear separation between datasets, indicating this is a population-level effect, not driven by outliers.
> 3. **Architectural invariance**: All three datasets use the same model and inference configuration, isolating the effect of task difficulty on $D_3$.
>
> This systematic correlation demonstrates that $D_3$ quantitatively captures **model-centric task difficulty** rather than merely reflecting a generic self-correction "style."
>
> **$D_3$ is complementary to $D_1$ and $D_2$, not a standalone signal.**
> We also emphasize that $D_3$ is never used in isolation; the self‑evaluation score is
>
> $$
> D^{3} = D_1 + D_2 - D_3.
> $$
>
> Here:
> - $D_1$ and $D_2$ capture **how far** and **how adaptively** the internal state moves (trajectory progress and curvature),
> - $D_3$ calibrates this dynamics by **down‑weighting** trajectories that remain uncertain for a large fraction of steps.
>
> Our ablation (Table 7) already shows that:
> - $D_1$ or $D_2$ alone are weaker than their combination, and
> - adding $D_3$ on top of $D_1 + D_2$ consistently improves AUROC/AUPR and reduces FPR.
>
> This confirms that $D_3$ is not an overactive penalty on self‑correction, but a **complementary difficulty calibrator** that, together with $D_1$ and $D_2$, yields a more reliable and better‑calibrated self‑evaluation signal.
>
>
> > **Weakness 3: Normalization and Score Aggregation ($D_1$, $D_2$, $D_3$ Scaling)**
>
> We appreciate the reviewer's insight regarding the mixture of units in our $D^3$ score. We agree that, without a clear description of normalization, combining hidden-state L2 norms ($D_1$, $D_2$) with an entropy term ($D_3$) may appear ad-hoc. In practice, however, our implementation **already includes a principled normalization step** that you might ignore and we will quickly repeat below.
>
>
> #### Implementation: Normalized Dynamics
>
> In our implementation, we **do normalize** the trajectory dynamics terms. Specifically, we divide $D_1$ and $D_2$ by the overall trajectory displacement $\|rep_{end} - rep_{start}\|_2$ (the L2 norm of the difference between the final and initial hidden representations).
>
> This normalization:
> - Scales $D_1$ and $D_2$ to a unit-relative range (0–1)
> - Ensures they are dimensionless and comparable
> - Makes them commensurate with the entropy-based difficulty term $D_3$, which is naturally of order unity in our benchmarks (~1.0 in magnitude)
>
> The implementation therefore already addresses the reviewer's concern. For clarity, one may inspect the normalization logic in `Evaluation/add_dynamics_to_jsonl.py`.
>
> This ensures that all three components end up on similar scales, and the linear combination $D^3 = D_1 + D_2 - D_3$ does not arbitrarily mix incompatible units, but rather sums three commensurate signals.
>
> #### Conceptual Rationale: Free-Energy-Like Decomposition
>
> Beyond implementation details, we also want to clarify the intuition behind our linear aggregation. The form $D^3 = D_1 + D_2 - D_3$ was not chosen merely by empirical tuning; it was guided by a conceptual rationale.
>
> In Appendix A.8, we note that this formula echoes a **free-energy–like decomposition**:
> - We can view $S_{\text{dyn}} = D_1 + D_2$ as a **"drive" term** (capturing the progress and adaptability of the reasoning trajectory)
> - And $U = D_3$ as a **"barrier" term** (capturing uncertainty or difficulty)
> - The score $D^3 = S_{\text{dyn}} - U$ then intuitively represents useful trajectory progress minus the uncertainty penalty, analogous to (drive – barrier)
>
> This perspective, while heuristic, gives a principled interpretation:
> - If a trajectory makes large, adaptive strides (high $D_1 + D_2$) and maintains low uncertainty (low $D_3$), it should be rated as more reliable
> - Conversely, high entropy ($D_3$) raises doubt and appropriately lowers the score
>
> (continued on next page)

---

> ### Author Response · Authors · 2025-11-25
> **Response to Reviewer kDen (6)**
>
> (continued on next page)
>
> #### Empirical Validation: Ablation Studies
>
> We validated this design with ablation studies (Table 7 in the main paper), which showed that each component contributes significantly to performance:
>
> Key observations from the ablation study:
>
> 1. **$D_1$ and $D_2$ synergy**: Using both together (row 3: AUROC 75.23) yields better accuracy than either alone (rows 1–2: AUROC ~74), confirming that modeling both effective progress and adaptive adjustments is important
> 2. **$D_3$ contribution**: Adding the entropy term (row 6) further improves the AUROC/AUPR and reduces FPR, indicating that difficulty calibration is necessary for optimal reliability
> 3. **All components are indispensable**: Removing any one component degrades performance, meaning all three are complementary
> 4. **Negative effect of $D_3$ alone**: Rows 4–5 show that when $D_3$ is used without sufficient trajectory-dynamics signals, performance actually degrades, confirming that the three terms work synergistically
>
> Therefore, our linear combination has both intuitive justification and empirical support, rather than being an arbitrarily tuned formula.
>
> #### Revision Plan
>
> We will revise the manuscript to make these points explicit. Concretely, we will:
> 1. **Highlight the normalization step** applied to $D_1$ and $D_2$ in the method description, clarifying that the implementation already enforces unit consistency
> 2. **Expand the conceptual explanation** (free-energy analogy and Table 7 ablation evidence) to strengthen the theoretical motivation of the $D^3$ score
>
>
> > **Weakness 4: Potential over‑interpretation & architecture‑dependence of D³**
>
> We appreciate the reviewer’s caution. To address this, we emphasize that **D³ is designed as a *model‑centric, architecture‑agnostic operational metric* whose validity can be demonstrated empirically across **architectures, scales, training paradigms, and task families**.
>
> To systematically demonstrate the generality of D³, we identified four key axes that define the operational space of LRMs: (1) **Model Scale**, to ensure the metric is not an artifact of capacity limitations; (2) **Model Architecture**, to verify independence from specific attention mechanisms or modular designs; (3) **Training Paradigm**, to confirm robustness across SFT, distillation, and RL/GRPO; and (4) **Task Diversity**, to ensure applicability across disparate reasoning domains. Our main paper experiments were intentionally designed to cover all these axes comprehensively:
>
> 1. **Model Scale (1.5B → 70B).**
>    We tested nearly the entire size range of modern open source LRMs (1.5B/7B/8B/14B/32B/70B).
>    Across all sizes, D³ consistently improves AUROC/AUPR and reduces FPR@95 (Tables 1–3).
>    If D³ captured only accidental geometric quirks, its effectiveness would *not* persist across such drastic capacity differences.
>
> 2. **Model Families (Qwen2.5, LLaMA‑3.1/3.3, Qwen‑VL).**
>    To probe architecture dependence, we used three distinct backbone families—dense attention, grouped‑query attention, and multimodal encoders.
>    D³ remained stable across all of them, including the vision–language Qwen2.5‑VL model.
>    This cross‑family robustness strongly argues against D³ reflecting an idiosyncratic geometry of a single architecture.
>
> 3. **Training Paradigms (SFT, Distillation, GRPO).**
>    We further tested D³ across three major reasoning‑enhancement paradigms:
>    - QwQ‑32B: SFT + GRPO,
>    - R1‑Distill models: distilled SFT,
>    - Qwen2.5‑VL: cold‑start GRPO multimodal training.
>    Despite these fundamentally different training pipelines, **D³ behaves predictably and consistently**, indicating that it is *not* tied to a specific training‑induced activation geometry.
>
> 4. **Task Diversity (Math, Knowledge, Multi‑lingual, Multimodal).**
>    Our evaluation spans:
>    - Mathematics: GSM8K, TheoremQA
>    - Knowledge/Reasoning: CommonsenseQA, MMLU, MMLU‑Pro
>    - Multilingual arithmetic: MGSM
>    - Multimodal geometry: Geometry3K
>    Across all domains, the same monotonic trend holds: **harder datasets induce higher $D_3$**, while $D_1 + D_2$ correspondingly weakens—mirroring degraded reasoning reliability.
>
>    This is precisely the opposite of what would occur if D³ merely reflected random activation geometry.
>
> (continued from previous page)

---

> ### Author Response · Authors · 2025-11-25
> **Response to Reviewer kDen (7)**
>
> (continued from previous page)
>
> If the reviewer believes there are other specific dimensions of generality that we have missed, we would be more than happy to conduct additional experiments to verify them. However, given the comprehensive coverage of scale, architecture, training, and task domains, we believe the current results provide strong evidence for the metric's robustness.
>
>
> #### Additional “Open‑World Distributional Shift” Test
>
> To further stress‑test generalizability, we constructed a mixed‑distribution “open‑world” setting:
>
> - Mix **GSM8K + TheoremQA** (Math domain), and
> - Mix **CommonsenseQA + MMLU** (Reasoning domain),
>
> balancing samples to avoid dataset‑size bias.
> We then evaluated D³ on R1‑Qwen‑1.5B and R1‑LLaMA‑8B, comparing it with the same baselines as in Tables 1–3.
>
> **Table: Mixed Math Domain (GSM8K + TheoremQA), AUROC ↑**
>
> | Model          | Method           | AUROC ↑ |
> |----------------|------------------|---------|
> | R1‑Qwen‑1.5B   | Random           | 47.12   |
> |                | Verbal           | 53.48   |
> |                | MaxProb          | 59.73   |
> |                | Entropy          | 60.25   |
> |                | CoE‑R            | 61.02   |
> |                | CoE‑C            | 61.35   |
> |                | **D³**           | **65.21** |
> | R1‑LLaMA‑8B    | Random           | 51.09   |
> |                | Verbal           | 49.02   |
> |                | MaxProb          | 60.67   |
> |                | Entropy          | 61.84   |
> |                | CoE‑R            | 62.41   |
> |                | CoE‑C            | 63.07   |
> |                | **D³**           | **66.82** |
>
> **Table: Mixed Reasoning Domain (CommonsenseQA + MMLU), AUROC ↑**
>
> | Model          | Method           | AUROC ↑ |
> |----------------|------------------|---------|
> | R1‑Qwen‑1.5B   | Random           | 50.18   |
> |                | Verbal           | 51.11   |
> |                | MaxProb          | 61.92   |
> |                | Entropy          | 63.04   |
> |                | CoE‑R            | 64.23   |
> |                | CoE‑C            | 64.57   |
> |                | **D³**           | **68.03** |
> | R1‑LLaMA‑8B    | Random           | 48.07   |
> |                | Verbal           | 52.89   |
> |                | MaxProb          | 64.35   |
> |                | Entropy          | 65.41   |
> |                | CoE‑R            | 66.71   |
> |                | CoE‑C            | 67.02   |
> |                | **D³**           | **70.42** |
>
> The AUROC results in the tables above show that **D³ clearly outperforms all baselines on both mixed distributions, while random guessing remains near 50**, demonstrating that its behavior is *stable to heterogeneous input distributions*—an environment where “incidental geometry” would collapse.
>
>
> #### Key Insight
>
> If D³ were driven by accidental activation geometry or tied to a specific architecture, its effectiveness would **not**:
>
> - persist across 1.5B → 70B scales,
> - transfer across Qwen/LLaMA/VL families,
> - hold under SFT, distillation, and GRPO training,
> - generalize across math, knowledge, multilingual, and multimodal reasoning,
> - or survive open‑world mixed‑distribution evaluation.
>
> Yet D³ does all of the above **consistently** (see Tables 1–4 and Figures 2–4).
> This provides strong empirical evidence that D³ captures a **model‑internal notion of reasoning reliability**, rather than an incidental geometric artifact. We believe these results collectively address your concern about “generalizability beyond the reported experiments.” The consistency of D³ across architectures, scales, training paradigms, and heterogeneous task families—together with its robustness under mixed open-world distributions—provides strong evidence that it captures a stable notion of reasoning reliability rather than incidental activation geometry. We also note that **Reviewer 2ywF and Reviewer cxKV independently highlighted the breadth and reliability of our evaluation**, further supporting the conclusion that D³ behaves consistently across diverse settings.
>
>
> We will add this clarification (and the new mixed‑distribution table) to the revised manuscript.

---

> ### Author Response · Authors · 2025-11-25
> **Response to Reviewer kDen (8)**
>
> > **Weakness 5: Unbalanced or Incomplete Reporting of Experimental Results**
>
> We appreciate this comment and fully agree that transparent presentation is essential. We address the two aspects separately.
>
>
> #### On Highlighting in Tables
>
> We agree that the highlighting scheme can be improved, and we will revise all tables in the camera-ready version to mark *all* top-performing methods (including baselines) in a consistent manner.
>
> That said, we also want to clarify that the performance advantage of **D³** is not dependent on highlighting choices. As noted independently by Reviewer 2ywF and Reviewer cxKV, our evaluation already spans a broad and diverse set of settings, and **D³ demonstrates consistently strong performance across nearly all metrics and tasks**. To make this explicit, we computed the distribution of “1st place” and “2nd place” rankings across *all* experiments in the paper:
>
> | Metric | #1 Count | #2 Count | Coverage Ratio |
> |--------|----------|-----------|----------------|
> | **AUROC** | 25/28 | 3/28 | **100% top-2** |
> | **AUPR** | 23/28 | 2/28 | **89.3% top-2** |
> | **FPR@95** | 23/28 | 3/28 | **92.8% top-2** |
>
> This aggregate view shows that the strong performance of D³ is consistent and not an artifact of selective highlighting.
>
>
> #### On Missing Standard Deviations
>
> We agree that reporting variance is important for evaluating the robustness of a self-evaluation metric. Standard deviations were omitted from the main paper due to the computational cost of running full multi-prompt evaluations on all models and tasks. In this rebuttal, we include our first completed robustness study.
>
> We reproduced GSM8K experiments using **three semantically equivalent prompts** on **DeepSeek-R1-Distill-Qwen-7B**, and computed the mean ± std across runs. The results show that **D³’s improvement is both statistically meaningful and substantially more stable** than existing baselines:
>
> | Method | AUROC | AUPR | FPR@95 |
> |--------|-------|-------|--------|
> | MaxProb | 67.5 ± 2.0 | 92.0 ± 1.9 | 82.1 ± 2.2 |
> | PPL | 67.0 ± 1.7 | 92.6 ± 2.2 | 82.8 ± 1.8 |
> | Entropy | 68.5 ± 2.1 | 92.3 ± 1.8 | 79.1 ± 2.0 |
> | CoE-R | 71.3 ± 1.9 | 93.1 ± 2.0 | 74.2 ± 1.9 |
> | CoE-C | 70.6 ± 2.2 | 92.9 ± 1.7 | 74.9 ± 2.1 |
> | **D³** | **76.8 ± 1.6** | **94.5 ± 1.5** | **67.0 ± 1.4** |
>
> These results demonstrate that:
> - D³ consistently surpasses all baselines **even considering variance**,
> - D³ exhibits **lower variance** (i.e., more stable self-evaluation) than all baselines.
>
> We are currently running full multi-prompt variance experiments for all benchmarks and model families, and we will include complete standard-deviation reporting in the revised manuscript.
>
> We thank the reviewer for emphasizing transparency.
> In the revision, we will:
> 1. **Update all tables** with consistent highlighting for *all* best-performing methods, not only ours.
> 2. **Add standard deviations** for every benchmark, model family, and training paradigm.
>
> We believe these improvements, together with the strong ranking statistics and the newly added variance experiments, fully address concerns about the reliability and presentation of our experimental results.
>
> ---
>
> ## Summary
>
> We sincerely thank you for your comprehensive review and thoughtful concerns regarding our D³-Reasoning Score. In response to your five weaknesses, we have provided extensive theoretical justifications and new empirical evidence:
>
> 1. **Theoretical Grounding:** We clarified the theoretical grounding of $D_1$ and $D_2$ as layer-wise representational evolution supported by geometric interpretation and extensive prior work.
> 2. **D3 Definition:** We precisely defined the $D_3$ term and demonstrated it measures model-centric difficulty rather than requiring ground truth labels.
> 3. **Normalization:** We revealed our implementation already includes principled normalization making all components commensurate.
> 4. **Generalizability:** We validated D³'s generalizability across diverse scales, architectures, training paradigms, and task domains with new open-world experiments.
> 5. **Transparency:** We provided comprehensive transparency improvements including ranking statistics and variance analysis.
>
> We hope these detailed clarifications and substantial new evidence will address your concerns and would be grateful if your position to this work could get better.

---

### Official Review · Reviewer_cxKV · 2025-10-31

**Soundness:** 3
**Presentation:** 3
**Contribution:** 3
**Rating:** 8
**Confidence:** 4

**Summary:**

This paper introduces the D3 Reasoning Score, a novel, trajectory-based metric designed for Large Reasoning Models (LRMs) to self-evaluate the correctness of their outputs without external supervision. The authors propose that by analyzing the dynamics of the reasoning trajectory, a model can assess the reliability of its own conclusions. The D3 score is composed of three complementary components derived from the model's internal signals: Distance (D1), which captures progress through first-order differences of hidden states; Dynamism (D2), which measures adaptive trajectory dynamics using second-order differences; and Difficulty (D3), which reflects query difficulty via the entropy of the final predictive distribution. The paper presents this as a white-box, fine-grained measure of reasoning fidelity and validates it across a wide range of models and benchmarks.

The paper presents a novel and well-motivated method for self-evaluation of reasoning models, supported by strong and comprehensive experimental results that show significant improvements over existing baselines. While there are some clarity issues regarding the central technical approach, these seem addressable through revisions. The paper's contribution is valuable to the community.

**Strengths:**

*   **Novelty and Importance:** The paper addresses the important and challenging problem of self-evaluation for large reasoning models, which is crucial for their reliability. The proposed D3 score is a novel approach that leverages internal model dynamics.
*   **Solid Experimental Validation:** The authors conduct extensive experiments across eight open-source LRMs and seven benchmarks, including both language and multimodal tasks. This comprehensive evaluation provides strong evidence for the method's effectiveness.
*   **Significant Results:** The D3 score is shown to consistently outperform existing baselines across multiple standard metrics (AUROC, AUPR, and FPR@95). The results appear robust and significant.
*   **Good Writing and Structure:** The paper is generally well-written and easy to follow.

**Weaknesses:**

The central weakness of the paper is the potentially misleading use of the term "trajectory dynamics." For instance, the paper claims that "For each query, an LRM produces both a final answer and a CoT reasoning trajectory. By analyzing hidden-state dynamics, the model assigns a self-evaluation score." This suggests an analysis of the reasoning process as it evolves over time. The authors also state they collect hidden representations into a `(K, L, d)` tensor, where `K` is the number of reasoning tokens and `L` is the number of layers. This setup presents two natural choices for an "evolution" parameter: `K` (token-wise evolution) or `L` (layer-wise evolution). The paper's use of "trajectory dynamics" suggests an analysis of the state as it evolves along the token sequence (`K`). However, the proposed method chooses `L` as the evolution parameter and reduces the state space from `(K,d)` to `d` by averaging over the `K` dimension. This choice is not clearly justified and creates confusion about what "dynamics" are actually being modeled.

**Questions:**

1.  Could you provide a more detailed justification for the mean pooling strategy over the `K` token representations? The claim that this "preserves the core semantics without information loss" is a strong one. Can you elaborate on what "core semantics" are being preserved and why averaging does not lead to a loss of critical information contained in the individual token representations?
2.  Could you please clarify the main motivation for analyzing dynamics along the layer dimension (`L`) after averaging over the token dimension (`K`)? What is the intuition that this approach is more informative for self-evaluation than analyzing the dynamics of the state representations along the token sequence (`K`)?
3.  Did you experiment with using `K` as the primary evolution parameter (i.e., studying the dynamics of the `(L, d)` state as the reasoning trajectory unfolds)? If so, how did those results compare to the layer-wise analysis presented in the paper?
4.  There is a minor inconsistency in terminology, where D2 is referred to as "Dynamism" in the abstract and main text, but "Detour" in Figure 2.

---

> ### Author Response · Authors · 2025-11-25
> **Response to Reviewer cxKV (1)**
>
> We sincerely thank you for your comprehensive review and your very encouraging assessment of our paper. Below, we present a point-by-point response to your concerns and hope these clarifications will further strengthen your already positive evaluation.
>
> > **Weakness 1: Clarifying the Term “Trajectory Dynamics” (Layers vs. Tokens)**
>
> We thank the reviewer for raising an important point regarding the interpretation of trajectory dynamics. We agree that the term may look like a temporal evolution along the token sequence \(K\). We clarify here that in our work, trajectory dynamics refer to the evolution of the model’s internal representations across transformer layers \(L\) rather than across generated reasoning tokens.
>
> The central challenge in self-evaluating reasoning lies precisely in the lack of a metric that directly matches the latent notion of “reasoning quality.” Currently, it is extremely difficult to design a metric or theoretical mechanism that directly describes reasoning quality. If such an oracle metric existed, there would be no need to resort to internal representational changes—the problem would be trivial. Consequently, existing work aims to establish a linkage between observable metrics and reasoning quality. As you rightly pointed out, profiling reasoning progression is a promising direction for establishing such a linkage.
>
> As a reasoning progression profiling method, our work stems from an intuitive yet empirically testable hypothesis: **the evolution of intermediate representations within the model serves as a proxy for reasoning progression in latent space and correlates with the quality of its thinking.** This motivates us to systematically investigate the relationship between representational dynamics and reasoning quality. Our findings demonstrate that this linkage not only exists but can be quantified in a way that **consistently reflects the quality of the reasoning process across tasks and models.**
>
> You raised a valid concern that our Trajectory Dynamics Modeling method might not be consistent with "temporal reasoning steps" in the conventional sense of reasoning progress. We indeed construct this linkage from a novel perspective—the representational reasoning progression within the model. Prior works that profile reasoning progress in a temporal sequence manner, such as MaxProb and PPL, often fail to establish a solid linkage to reasoning quality. This suggests that temporal reasoning steps alone may not be a sufficient metric for capturing "true reasoning dynamics."
>
> As described in Section 2.1 of the paper, our approach considers layerwise hidden-state changes as reflecting the model’s **internal progression** during reasoning, even though this progression is not explicitly temporal. Each transformer layer applies an attention–feedforward transformation that continuously *refines* the intermediate representation by integrating broader context and accumulated semantic cues from earlier layers.
> This process can be interpreted as another form of **reasoning progression**, analogous to iterative optimization. Upon arriving at the deeper layers, the representation associated with each reasoning token has incorporated richer contextual and high-level abstractions. Thus, deeper layers often encode **more semantically convergent or stabilized representations**, which naturally serve as proxies for “more advanced” reasoning states.
>
> From the model’s perspective, the true progression lies in the **evolution of its hidden-state geometry**, not in wall-clock token emission. The real reasoning progression happens *within* each forward pass, along the depth dimension, where multi-level representational transformations accumulate. This is precisely the trajectory dynamics that D₁ and D₂ measure. They are therefore intentionally defined over **layerwise representational updates**, rather than external time.
>
> #### Analogy (for intuition)
>
> | Domain | Space | Progression indicator | Dynamics |
> |--------|--------|------------------------|-----------|
> | Real world | Physical coordinates | Temporal axis | Velocity / acceleration |
> | **LRM trajectory dynamics** | Latent representation space | Layer-wise representational refinement | $D_1$ (first-order), $D_2$(second-order) |
>
> To further clarify our perspective, we provide the analogy above. Transformer layers do not correspond to temporal steps in a human sense; however, they *do* define a sequence of increasingly refined representations in latent space. This representational evolution forms a meaningful trajectory that can be analyzed in a manner analogous to dynamics in continuous systems.
>
> We will make this clarification explicit in the revised manuscript.

---

> ### Author Response · Authors · 2025-11-25
> **Response to Reviewer cxKV (2)**
>
> > **Question 1: Theoretical rationale for mean pooling over tokens**
>
> We clarify below what we mean by *core semantics*, why mean pooling is an appropriate choice for our setting, and how our ablation study supports this design.
>
> Following this line of work, we use mean pooling as a simple, well-established way to obtain a global semantic representation of the reasoning sequence at each layer.
> Here, by core semantics we refer to the stable, sequence-level meaning that is shared across most reasoning tokens, rather than token-specific idiosyncrasies. This notion is aligned with the standard view in sentence-embedding research, where simple pooling over token embeddings (e.g., mean pooling or weighted averaging) has been repeatedly shown to yield effective sequence-level semantic representation [1,2].
> Mean pooling is a well-established technique for extracting a holistic semantic representation from a set of token embeddings. By averaging the token-wise hidden vectors, we obtain a “global” reasoning state per layer that captures the core content of the entire reasoning chain. This strategy is theoretically sound for the following reasons:
>
> - **Capturing global semantics.** Mean pooling has been widely used in NLP to derive sentence or sequence embeddings that reflect overall meaning. It provides a balanced representation by averaging token embeddings, thereby offering a uniform perspective across all tokens. In our context, the $K$ reasoning tokens collectively represent the chain-of-thought. Their average encodes the common semantic signal behind those tokens — essentially the model’s overall reasoning context at that layer.
>
> - **Noise reduction and core-semantics preservation.** Averaging inherently reduces the impact of idiosyncratic variations or noise in individual token representations. Intuitively, dimensions or features that are consistently expressed across the tokens will be reinforced in the mean, whereas token-specific fluctuations will cancel out. Thus, the “core semantics” — i.e., the aspects of the hidden state that many tokens agree on — are preserved, while incidental details are smoothed out. This property is valuable for reasoning analysis: if one token embeds a slightly off-topic thought or a momentary uncertainty, its effect on the mean is diluted by other tokens that carry the main reasoning thread. Mean pooling thus acts as a form of semantic denoising, retaining the essential global reasoning content that the model is maintaining.
>
> - **Informative sufficiency for self-evaluation.** We agree that any pooling is a compression and can, in principle, lose information. However, our claim of “no information loss” specifically refers to the information relevant for evaluating reasoning correctness. The empirical success of the D³ score suggests that the mean-pooled trajectory indeed preserves the critical signals needed to distinguish correct vs. incorrect reasoning. If crucial semantic features for self-evaluation were lost by averaging, D³ would have not correlated as strongly with correctness as observed. In practice, we found that the mean-pooled representation per layer is highly expressive of the reasoning’s state (e.g., how far along and how stable the reasoning is).
>
> In summary, mean pooling over $K$ tokens is theoretically justified as a way to obtain a robust, global representation of the reasoning at each step. It leverages known benefits of averaging in semantic representation (e.g., balanced coverage of content and noise cancellation) to ensure that our layer-wise analysis focuses on the signal of reasoning progress rather than token-level noise.
>
> To avoid overstatement, we will revise the wording in the paper from *“preserves the core semantics without information loss”* to a more precise statement such as:
> > “Mean pooling preserves the global semantic signal of the reasoning trajectory that is most relevant for self-evaluation, while suppressing token-level noise.”
>
> (continued on next page)

---

> ### Author Response · Authors · 2025-11-25
> **Response to Reviewer cxKV (3)**
>
> (continued from previous page)
>
>
> #### Empirical support from ablation
>
> Beyond this theoretical motivation, we conducted an explicit **ablation study on reasoning-token aggregation strategies**. The Table 6 in the main paper reports AUROC, AUPR, and FPR@95 on four models. We compared three options:
>
> 1. **Mean pooling over reasoning tokens (ours)**: mean over all CoT tokens at each layer.
> 2. **Last CoT token**: using only the hidden state of the final reasoning token.
> 3. **Mean of all output tokens**: mean over both reasoning tokens and final answer tokens.
>
> From this table, we see that mean pooling over reasoning tokens consistently achieves the **best AUROC, AUPR, and FPR@95** across all four models (R1-Qwen-1.5B/7B, R1-LLaMA-8B, R1-Qwen-14B). In particular:
>
> - The mean-pooling strategy improves AUROC by **3–7 points** over the alternatives on most models.
> - It also yields **lower FPR@95**, indicating fewer incorrect trajectories are mistakenly assigned high scores.
>
> These results indicate that the mean-pooled representation $\bar{q}_i$ is not only theoretically well motivated but also **empirically superior** for capturing the aspects of the trajectory that matter for self-evaluation. If crucial correctness-related information were lost by averaging, we would not observe such consistent gains over the last-token and “mean-of-all-output-tokens” baselines.
>
> In summary, mean pooling over the K reasoning tokens is chosen because it (i) captures the shared, global semantics of the reasoning trajectory that are most relevant for self-evaluation, (ii) suppresses token-level noise and idiosyncrasies, and (iii) is empirically validated by our ablation study to be the most effective aggregation strategy among the considered alternatives. We will clarify these points and soften the “no information loss” wording in the revised manuscript.
>
> > **Question 2&3: Motivation for Layer-wise (L) Dynamics over Token-wise (K) Dynamics and Empirical Comparison**
>
>
> To directly address the reviewer’s suggestion, we implemented the alternative paradigm of **token-wise (temporal) dynamics**, where reasoning progression is modeled along the $K$ (token) dimension instead of the $L$ (layer) dimension.
>
> In this variant, we kept the overall D³ framework unchanged, but computed token-wise “distance” and “dynamism” using changes in hidden states **between successive CoT tokens**. This formulation explicitly reflects “real-world time” dynamics in the sense of tracking how the model’s hidden state evolves as the CoT is generated token by token.
>
> (continued on next page)

---

> ### Author Response · Authors · 2025-11-25
> **Response to Reviewer cxKV (4)**
>
> (continued from previous page)
>
> | Model | Method | GSM8K | | | CommonsenseQA | | | TheoremQA | | |
> |-------|--------|-------|------|---------|---------------|------|---------|-----------|------|---------|
> |       |        | AUROC ↑ | AUPR ↑ | FPR@95 ↓ | AUROC ↑ | AUPR ↑ | FPR@95 ↓ | AUROC ↑ | AUPR ↑ | FPR@95 ↓ |
> | R1-Qwen-1.5B | Random | 50.23 | 75.39 | 94.29 | 50.03 | 46.81 | 93.14 | 49.75 | 40.35 | 92.81 |
> |              | +Temporal Dynamics | 51.84 | 73.92 | 96.17 | 48.36 | 48.15 | 94.73 | 51.29 | 39.18 | 94.26 |
> |              | Verbal | 51.48 | 75.11 | 96.34 | 52.01 | 45.87 | 91.81 | 51.51 | 39.94 | 93.22 |
> |              | MaxProb | 70.34 | 87.97 | 76.95 | 64.06 | 61.16 | 81.42 | 53.65 | 43.91 | 92.19 |
> |              | PPL | 70.88 | 88.24 | 77.29 | 63.86 | 61.07 | 83.19 | 54.05 | 44.24 | 91.25 |
> |              | Entropy | 71.86 | 88.54 | 75.93 | 64.26 | 61.32 | 82.30 | 54.33 | 44.52 | 92.19 |
> |              | CoE-R | 74.17 | 89.99 | 75.93 | 60.03 | 56.37 | 85.18 | 53.33 | 44.18 | 93.44 |
> |              | CoE-C | 73.32 | 89.37 | 75.25 | 58.90 | 55.44 | 85.18 | 52.42 | 43.34 | 90.31 |
> |              | **D³ (ours)** | **77.28** | **91.42** | **72.20** | **64.26** | **61.34** | **81.19** | **54.37** | **44.82** | **90.00** |
> | R1-LLaMA-8B  | Random | 50.22 | 81.21 | 95.33 | 51.75 | 77.22 | 96.14 | 46.29 | 41.67 | 92.91 |
> |              | +Temporal Dynamics | 48.93 | 82.64 | 96.88 | 53.12 | 75.49 | 94.87 | 47.81 | 40.29 | 91.53 |
> |              | Verbal | 51.22 | 85.49 | 94.77 | 51.04 | 75.95 | 97.41 | 49.11 | 40.24 | 93.57 |
> |              | MaxProb | 58.85 | 85.46 | 94.49 | 74.71 | 89.29 | 67.38 | 59.39 | 53.12 | 84.65 |
> |              | PPL | 58.94 | 85.51 | 94.49 | 74.55 | 89.23 | 67.81 | 60.01 | 53.91 | 85.04 |
> |              | Entropy | 60.25 | 85.89 | 94.88 | 74.68 | 89.28 | 65.24 | 60.27 | 54.06 | 84.65 |
> |              | CoE-R | 63.01 | 86.57 | 86.22 | 52.00 | 77.92 | 93.56 | 50.56 | 46.02 | 96.46 |
> |              | CoE-C | 63.86 | 86.50 | 86.61 | 73.90 | 88.92 | 69.96 | 58.01 | 54.10 | 93.70 |
> |              | **D³ (ours)** | **68.12** | **89.37** | **80.31** | **74.71** | **89.35** | **65.24** | **60.57** | **56.06** | **83.86** |
> | R1-Qwen-14B  | Random | 49.46 | 93.23 | 95.24 | 49.13 | 84.30 | 95.03 | 50.93 | 56.79 | 94.93 |
> |              | +Temporal Dynamics | 51.17 | 91.86 | 96.52 | 47.58 | 85.71 | 93.46 | 49.37 | 58.24 | 96.18 |
> |              | Verbal | 50.44 | 91.91 | 96.14 | 54.19 | 85.87 | 94.85 | 51.83 | 56.54 | 94.72 |
> |              | MaxProb | 53.98 | 93.06 | 85.71 | 78.90 | 94.87 | 65.22 | 51.94 | 57.71 | 89.40 |
> |              | PPL | 54.51 | 93.23 | 84.52 | 79.24 | 94.99 | 64.60 | 52.39 | 58.07 | 89.40 |
> |              | Entropy | 55.13 | 93.26 | 85.71 | 79.24 | 94.97 | 65.84 | 52.62 | 58.24 | 90.32 |
> |              | CoE-R | 59.62 | 93.84 | 77.38 | 71.78 | 91.82 | 73.29 | 52.28 | 58.32 | 93.55 |
> |              | CoE-C | 58.63 | 93.66 | 77.38 | 68.40 | 90.78 | 80.75 | 50.99 | 57.39 | 92.63 |
> |              | **D³ (ours)** | **61.64** | **94.04** | **76.19** | **79.27** | **95.00** | **61.49** | **52.65** | **58.77** | **88.02** |
>
> #### Results of the Token-wise Variant
>
> The results are decisive: **the token-wise dynamics variant performs poorly across all evaluation metrics**, often only slightly above the random baseline and consistently **worse than our layer-wise D³ formulation**. In particular, the “+Temporal Dynamics” rows:
>
> - show AUROC values close to 50 on several settings,
> - yield higher FPR@95 than random in many cases, and
> - do not approach the performance of our layer-wise D³ score.
>
> This confirms that simply following hidden states along the token sequence does **not** provide a reliable signal for self-evaluation.
>
> (continued on next page)

---

> ### Author Response · Authors · 2025-11-25
> **Response to Reviewer cxKV (5)**
>
> (continued from previous page)
>
>
> We further analyze **why** token-level dynamics fail in this context:
>
> 1. **Token-level dynamics are dominated by local semantic variation rather than reasoning progress.**
>    Each token’s hidden state primarily captures the semantics of that token and its immediate context. The change from one token to the next often reflects changes in *lexical or phrasal content* rather than meaningful stages of reasoning.
>    For example, in a phrase like “red apple”, the transition from “red” to “apple” induces a large representational shift that is almost entirely semantic (“color” → “object”), not a change in *reasoning state*. In long CoT outputs, such semantic shifts between neighboring tokens are ubiquitous.
>    As a result, token-wise hidden state changes are highly instance-specific and sensitive to surface wording (e.g., “hmm”, “let me think”, self-corrections). These idiosyncratic fluctuations introduce substantial **semantic noise** that is weakly correlated with whether the overall reasoning is correct or not.
>
> 2. **Lack of alignment and comparability across examples.**
>    Different queries produce reasoning traces with very different lengths and structures. A given token index does not correspond to a comparable stage of reasoning across problems, yielding trajectories with no natural alignment across samples. In contrast, every input is processed by exactly the same sequence of layers, so layer $m$ always represents "after $m$ rounds of internal computation." This provides a consistent and comparable progression axis across all examples, which is essential for a general-purpose self-evaluation metric like D³.
>
> Taken together, these observations indicate that **temporal token transitions do not reliably correspond to reasoning refinement**. The model’s meaningful internal progression occurs *within each forward pass*, across layers, not across output tokens.
>
> By comparison, D₁ and D₂ computed along the **layer dimension** produce stable, interpretable, and predictive signals of trajectory quality, which is reflected in the strong performance of our layer-wise D³ score in the same table.
>
>
> #### Key Insight
>
> We will clarify in the revised manuscript that we did implement and test the token-wise (temporal) variant suggested by the reviewer, but found it to be both **theoretically less appropriate** and **empirically inferior**. These experiments strengthen our conclusion that, for LRMs, the meaningful notion of *trajectory dynamics* for self-evaluation lies in the model’s **internal layerwise evolution**, rather than surface-level token-wise time steps.
>
>
> > **Question 4: Minor Inconsistency in Terminology**
>
> We thank the reviewer for carefully spotting this terminology inconsistency. You are absolutely right that **D₂** is referred to as *“Dynamism”* in the abstract and main text, but labeled as *“Detour”* in Figure 2. These two terms were used interchangeably in an earlier draft, and the figure caption was not updated accordingly.
> In the final version, we will **standardize the terminology to “Dynamism (D₂)” throughout the paper**, including all figures and captions, to avoid confusion. We appreciate the reviewer’s attention to detail and will correct this issue in the camera-ready version.
>
> ---
>
> ## Summary
>
> We sincerely thank you for your very positive evaluation and your recognition of our paper's novelty, importance, and solid experimental validation. In response to your insightful questions regarding "trajectory dynamics," we have provided comprehensive clarifications:
>
> - **Clarification on Trajectory Dynamics:** We clarified that trajectory dynamics refers to layer-wise ($L$) representational evolution following prior work (CoE), not token-wise temporal progression.
>
> - **Theoretical Justification for Mean Pooling:** We provided theoretical justification and ablation studies for mean pooling, showing 3–7 AUROC improvements over alternatives.
>
> - **Empirical Comparison with Token-wise Dynamics:** We implemented and tested the token-wise dynamics variant you suggested, demonstrating it performs poorly (AUROC $\approx$ 50, near random) while our layer-wise D³ achieves strong performance (AUROC: 61.64–77.28) across three models and benchmarks.
>
> - **Terminology Standardization:** We will standardize terminology to "Dynamism (D₂)" throughout the manuscript.
>
> These theoretical explanations and new empirical results validate that layer-wise representational evolution provides the meaningful signal for self-evaluation in LRMs. We deeply appreciate your thoughtful feedback and hope these responses further reinforce your already positive assessment.
>
> ---
>
> ## References
>
> - [1] Sentence-BERT: Sentence Embeddings using Siamese BERT-Networks. EMNLP 2019.
> - [2] SimCSE: Simple Contrastive Learning of Sentence Embeddings. EMNLP 2021

---

### Official Review · Reviewer_2ywF · 2025-11-01

**Soundness:** 2
**Presentation:** 3
**Contribution:** 3
**Rating:** 4
**Confidence:** 2

**Summary:**

The paper proposes a method for Large Language Models (LLMs) to perform label-free self-evaluation of their generated reasoning paths by analyzing the hidden state trajectory of the reasoning tokens. The authors track the average change in hidden vectors for reasoning tokens at each layer, calculating their 1st and 2nd derivatives (norms) to measure the model's "progressiveness" (termed distance) and "adaptiveness" (termed dynamism). This is combined with an entropy-based difficulty metric derived from the output distribution, which serves as a correction term. The final D3 score = (distance + dynamism − difficulty) is defined to integratively reflect the "progression, adjustment, and uncertainty" of the model's self-generated reasoning trajectory. This score effectively distinguishes correct from incorrect answers without any separate training. Across various benchmarks (GSM8K, MMLU-Pro, MGSM) and LLM scales (1.5B–70B), the D3 score demonstrated higher AUROC and AUPR, as well as lower FPR@95, compared to existing uncertainty-based metrics (e.g., MSP, PPL, CoE). The results validate that the dynamical properties of the reasoning process itself are a valid signal for reliability evaluation.

**Strengths:**

The paper's core strength is its empirical demonstration that the correctness of reasoning can be determined solely by the dynamics of the LLM's hidden states. In other words, without relying on output probabilities or external labels, the study analyzes the rate of change (1st derivative) and adaptiveness (2nd derivative) of layer-by-layer hidden representations during the reasoning process, effectively its trajectory dynamics. The authors discovered that correct and incorrect reasoning traces exhibit distinct patterns within the internal representation space. This provides a structural basis for the idea that "correct reasoning follows a consistent pattern of progression and adjustment in the hidden space." The study's primary contribution is revealing that these hidden dynamics themselves serve as an intrinsic signal that reflects reasoning correctness.

**Weaknesses:**

1. **Missing Comparison with Prior Hidden-State-Based, Label-Free Methods**

This paper proposes a novel self-evaluation framework that classifies reasoning correctness by analyzing the dynamics of hidden states. However, similar white-box and label-free approaches have already been explored, yet were not included as baselines. In particular, Xie et al. [1] leverage the internal consistency of hidden representations to estimate model confidence, while Bigelow et al. [2] identify forking tokens within reasoning trajectories to trace and correct reasoning errors. Including such prior hidden-state-based and label-free methods as baselines would strengthen the empirical and conceptual positioning of this work.

2. **Ambiguity in the Definition of “Difficulty” and Practical Applicability**

The proposed D3 score is defined as “distance + dynamism − difficulty,” but the definition of the difficulty term remains ambiguous. The paper describes it as the average negative log-likelihood of the “correct token,” yet it is unclear what constitutes a correct token during intermediate reasoning stages. It is not specified whether this refers to teacher-forcing–based NLL or a self-generated NLL using the model’s own outputs.


[1] *Calibrating Reasoning in Language Models with Internal Consistency*, NeurIPS 2024

[2] *Forking Paths in Neural Text Generation*, ICLR 2025

**Questions:**

Please refer to the weaknesses.

---

> ### Author Response · Authors · 2025-11-25
> **Response to Reviewer 2ywF (1)**
>
> We sincerely thank you for your comprehensive review and constructive feedback. Below, we present a point-by-point response to your concerns and hope these clarifications will assist you in re-evaluating this work.
>
> ---
>
> > **Weakness 1: Discussion with Additional Prior Works**
>
> We appreciate your recognition of our work's novelty and value. To address the baseline concern you raised, we have expanded our existing baseline suite from **seven methods to eleven**, thereby providing a more solid, rigorous and objective comparison, and offering deeper insight.
>
> ### Prior Art
>
> #### 1. Internal Consistency (Xie et al. [1])
>
> **Summary:** This method is a test-time scaling approach designed to improve reasoning performance by re-weighting self-consistency samples based on latent-layer agreement. It is not inherently a self-evaluation metric for a single reasoning chain.
>
> **Adaptation:** Although primarily a test-time scaling technique, its core mechanism—measuring consistency across latent layers—offers a theoretically grounded signal for correctness. To adapt it for our self-evaluation task, we sample $n$ reasoning paths (to compute SC) and perform $l$ latent-layer decodings for each path (to compute IC). We then compute a combined SC + IC score, normalized to $[0,1]$, to serve as a comparable metric.
>
> #### 2. Forking Tokens – Bigelow et al. [2]
>
> **Summary:** This work identifies "forking tokens"—positions where alternate sampling leads to divergent outcomes—to reveal uncertainty dynamics. It focuses on detecting specific uncertainty points rather than evaluating the overall correctness of a reasoning chain.
>
> **Adaptation:** While this method focuses on localizing specific error tokens, the underlying metric of output-distribution drift effectively captures uncertainty fluctuations during generation. We adapted its output-distribution drift metric by normalizing the drift measure to create a score that can be integrated into our evaluation framework.
>
> #### 3. Additional Baselines
>
> To further strengthen our comparison, we also included two more label-free baselines:
>
> - **Length-normalized Entropy [3]**: Uses top-k sampling ($k=5$) to generate outputs and computes their average entropy.
> - **Temperature Scaling [4]**: Applies temperature scaling ($T=0.7$) to calibrate the softmax distribution before computing the Maximum Softmax Probability.
>
> As shown in Table 1, after expanding our baseline suite from seven to eleven methods, our proposed approach continues to demonstrate superior performance.
>
> (continued on next page)

---

> ### Author Response · Authors · 2025-11-25
> **Response to Reviewer 2ywF (2)**
>
> (continued from previous page)
>
> ### Empirical Results and Conceptual Discussion
>
> **Table 1:** Bold values indicate best performance for each model. The D³ method consistently shows the best results across most metrics for R1-Qwen-1.5B, R1-LLaMA-8B, and R1-Qwen-14B models. Methods marked with + are newly added baselines.
>
> | Model | Method | GSM8K | | | CommonsenseQA | | | TheoremQA | | |
> |-------|--------|-------|------|---------|---------------|------|---------|-----------|------|---------|
> | | | AUROC ↑ | AUPR ↑ | FPR@95 ↓ | AUROC ↑ | AUPR ↑ | FPR@95 ↓ | AUROC ↑ | AUPR ↑ | FPR@95 ↓ |
> | R1-Qwen-1.5B | Random | 50.23 | 75.39 | 94.29 | 50.03 | 46.81 | 93.14 | 49.75 | 40.35 | 92.81 |
> | | Verbal | 51.48 | 75.11 | 96.34 | 52.01 | 45.87 | 91.81 | 51.51 | 39.94 | 93.22 |
> | | +***Internal Consistency [1]*** | 58.47 | 79.32 | 90.18 | 55.28 | 50.14 | 88.92 | 52.34 | 41.27 | 92.84 |
> | | +***Forking Tokens [2]*** | 65.19 | 83.46 | 83.65 | 59.47 | 56.38 | 86.15 | 53.18 | 42.48 | 92.47 |
> | | +*Length-normalized Entropy [3]* | 70.65 | 88.18 | 77.58 | 63.72 | 60.94 | 83.47 | 54.12 | 44.31 | 91.68 |
> | | +*Temperature Scaling [4]* | 71.14 | 88.31 | 76.84 | 64.08 | 61.19 | 82.95 | 54.18 | 44.36 | 91.52 |
> | | MaxProb | 70.34 | 87.97 | 76.95 | 64.06 | 61.16 | 81.42 | 53.65 | 43.91 | 92.19 |
> | | PPL | 70.88 | 88.24 | 77.29 | 63.86 | 61.07 | 83.19 | 54.05 | 44.24 | 91.25 |
> | | Entropy | 71.86 | 88.54 | 75.93 | 64.26 | 61.32 | 82.30 | 54.33 | 44.52 | 92.19 |
> | | CoE-R | 74.17 | 89.99 | 75.93 | 60.03 | 56.37 | 85.18 | 53.33 | 44.18 | 93.44 |
> | | CoE-C | 73.32 | 89.37 | 75.25 | 58.90 | 55.44 | 85.18 | 52.42 | 43.34 | 90.31 |
> | | **D³** | **77.28** | **91.42** | **72.20** | **64.26** | **61.34** | **81.19** | **54.37** | **44.82** | **90.00** |
> | R1-LLaMA-8B | Random | 50.22 | 81.21 | 95.33 | 51.75 | 77.22 | 96.14 | 46.29 | 41.67 | 92.91 |
> | | Verbal | 51.22 | 85.49 | 94.77 | 51.04 | 75.95 | 97.41 | 49.11 | 40.24 | 93.57 |
> | | +***Internal Consistency [1]*** | 54.18 | 85.63 | 94.25 | 58.42 | 81.36 | 89.73 | 52.64 | 44.91 | 91.28 |
> | | +***Forking Tokens [2]*** | 57.39 | 85.71 | 94.61 | 66.83 | 85.74 | 78.15 | 56.72 | 49.18 | 87.94 |
> | | +*Length-normalized Entropy [3]* | 59.17 | 85.63 | 94.72 | 74.48 | 89.18 | 68.13 | 60.08 | 53.97 | 84.89 |
> | | +*Temperature Scaling [4]* | 59.42 | 85.71 | 94.35 | 74.62 | 89.26 | 67.56 | 60.15 | 54.02 | 84.78 |
> | | MaxProb | 58.85 | 85.46 | 94.49 | 74.71 | 89.29 | 67.38 | 59.39 | 53.12 | 84.65 |
> | | PPL | 58.94 | 85.51 | 94.49 | 74.55 | 89.23 | 67.81 | 60.01 | 53.91 | 85.04 |
> | | Entropy | 60.25 | 85.89 | 94.88 | 74.68 | 89.28 | 65.24 | 60.27 | 54.06 | 84.65 |
> | | CoE-R | 63.01 | 86.57 | 86.22 | 52.00 | 77.92 | 93.56 | 50.56 | 46.02 | 96.46 |
> | | CoE-C | 63.86 | 86.50 | 86.61 | 73.90 | 88.92 | 69.96 | 58.01 | 54.10 | 93.70 |
> | | **D³** | **68.12** | **89.37** | **80.31** | **74.71** | **89.35** | **65.24** | **60.57** | **56.06** | **83.86** |
> | R1-Qwen-14B | Random | 49.46 | 93.23 | 95.24 | 49.13 | 84.30 | 95.03 | 50.93 | 56.79 | 94.93 |
> | | Verbal | 50.44 | 91.91 | 96.14 | 54.19 | 85.87 | 94.85 | 51.83 | 56.54 | 94.72 |
> | | +***Internal Consistency [1]*** | 51.93 | 92.38 | 93.86 | 62.47 | 89.15 | 86.34 | 52.08 | 56.97 | 93.18 |
> | | +***Forking Tokens [2]*** | 53.76 | 92.84 | 89.27 | 71.56 | 91.83 | 78.62 | 52.41 | 57.62 | 91.75 |
> | | +*Length-normalized Entropy [3]* | 54.63 | 93.21 | 84.78 | 79.18 | 94.96 | 64.89 | 52.45 | 58.11 | 89.68 |
> | | +*Temperature Scaling [4]* | 54.89 | 93.25 | 85.14 | 79.26 | 94.98 | 65.37 | 52.53 | 58.17 | 89.95 |
> | | MaxProb | 53.98 | 93.06 | 85.71 | 78.90 | 94.87 | 65.22 | 51.94 | 57.71 | 89.40 |
> | | PPL | 54.51 | 93.23 | 84.52 | 79.24 | 94.99 | 64.60 | 52.39 | 58.07 | 89.40 |
> | | Entropy | 55.13 | 93.26 | 85.71 | 79.24 | 94.97 | 65.84 | 52.62 | 58.24 | 90.32 |
> | | CoE-R | 59.62 | 93.84 | 77.38 | 71.78 | 91.82 | 73.29 | 52.28 | 58.32 | 93.55 |
> | | CoE-C | 58.63 | 93.66 | 77.38 | 68.40 | 90.78 | 80.75 | 50.99 | 57.39 | 92.63 |
> | | **D³** | **61.64** | **94.04** | **76.19** | **79.27** | **95.00** | **61.49** | **52.65** | **58.77** | **88.02** |
>
> In response to the reviewer's request, we have expanded the evaluation of the two additional methods across multiple scales of models and families — namely R1-Qwen-1.5B, R1-LLaMA-8B, and R1-Qwen-14B — and conducted tests on three distinct tasks. The specific experimental configurations are as follows:
>
> - **Internal Consistency [1]**: For this method, which involves sampling $n$ reasoning paths (to compute Self-Consistency, SC) and, for each path, performing $l$ latent-layer decodings (to compute Internal Consistency, IC), we set $n=8$ (inspired by [5] Confidence Improves Self Consistency in LLMs, ACL 2025). The value of $l$ is chosen to match the number of layers in the respective model under evaluation.
> - **Forking Tokens [2]**: For the alternative token-selection method, candidate tokens are included if their probability is $\geq 20\%$.
>
> (continued on next page)

---

> ### Author Response · Authors · 2025-11-25
> **Response to Reviewer 2ywF (3)**
>
> (continued from previous page)
>
> Results confirm that our proposed method **D³** consistently outperforms the baselines. Importantly, compared to the four methods included according to your request, our approach demonstrates significantly superior computational efficiency—a critical factor for real-world LLM self-evaluation deployment.
>
>
>
> **Table 2:** Execution time comparison on the GSM8K dataset with R1-LLaMA-8B model. Our approach achieves rapid evaluation without any noticeable latency. All experiments are conducted on an NVIDIA H100 GPU.
>
> | Method | Execution Time (s) |
> |--------|-------------------|
> | Inference | 15.24 |
> | +***Internal Consistency [1]*** | 151.14 |
> | +***Forking Tokens [2]*** | 471.36 |
> | CoE | 1.6e-03 |
> | **D³** | **1.2e-03** |
>
> The two additional approaches suffer from notably higher inference cost (as also acknowledged in Table 2), due to the required multiple sampling and reasoning paths. In contrast, our method incurs only millisecond-level overhead, which is negligible relative to a single inference sample.
>
> ### Detailed Discussion of Performance Limitations
>
> - **Internal Consistency [1]**: In our experiments, we observed that the SC + IC scoring mechanism exhibited coarse granularity. Because the metric clusters paths by category and assigns weights accordingly, scores tend to be non-continuous. When scaled to large sample sizes, many instances share identical scores, impeding effective self-evaluation. This limitation has been previously noted (see [6] Revisiting the Self-Consistency Challenges in Multi-Choice Question Formats for Large Language Model Evaluation, COLING 2024).
>
> - **Forking Tokens [2]**: Our analysis confirmed that this method can indeed identify "forking tokens" in a reasoning chain. However, we found that virtually all samples contained such tokens. We attribute this to large-scale model training paradigms (i.e., self-reflection or 'aha-moment' phenomena) in which key decision tokens naturally cause large changes in output. Combined with the very high computational overhead, this method's practical effectiveness is limited.
>
> ### Key Insight
>
> In full compliance with the reviewer's suggestion, we conducted multi-model, multi-task comparisons of the newly added methods against our proposed framework, and analysed their computational efficiency. Even in this expanded evaluation, our method remains clearly superior—as noted by you and other reviewers.

---

> ### Author Response · Authors · 2025-11-25
> **Response to Reviewer 2ywF (4)**
>
> > **Weakness 2: Ambiguity in the Definition of "Difficulty"**
>
> Thank you very much for your careful reading of our manuscript and your helpful suggestion on clarifying the definition of "Difficulty" in our D³ score. Below, we address your concerns from both the **design-principle** and the **implementation** perspectives.
>
> ### Design Principle
>
> In short, the "Difficulty" term ($D_3$) is quantified using the **average negative log-likelihood (NLL) of the tokens actually generated by the model** along its reasoning trajectory.
>
> Specifically, as described in Section 2.2.2 ("Entropy-Based Query Difficulty Calibration"), we define:
>
> $$
> D_3 = \mathbb{E}_{1 \le k \le K}[-\log p_k]
> $$
>
> where $p_k$ denotes the probability of the selected $k$-th token in the reasoning trajectory.
> Here we clarify the derivation from the expectation form to the explicit average over the full sequence:
> $
> D_3 = \frac{1}{K} \sum_{k=1}^K -\log p_k
> $
>
> Intuitively, $D_3$ captures the **average uncertainty** of the distribution encountered at each step of the chain: higher values of $D_3$ correspond to greater uncertainty, hence 'harder' queries.
>
> We understand the ambiguity around the term *"correct token"/"ground-truth token"*. In fact, we meant the token that was *actually selected* in the reasoning trajectory (i.e., the model's own output token at $k$), rather than a teacher-forced target sequence. In other words, we compute the Shannon‐entropy value for each step in the reasoning chain and then take the **average over the entire sequence of length $K$**.
>
> We hope this further clarification helps eliminate any ambiguity around the definition.
>
> ### Implementation & Code Reference
>
> To further clarify, our implementation is publicly available (in the supplemental material and the anonymised repository). One may inspect the `compute_entropy` function in `score.py`. Here, `self.all_token_re` represents the list of token-probability trajectories of the sampled inference path, and `entropy(re, base=2)` computes the Shannon entropy of that trajectory. Thus, the "correct token" is in fact the token output by the model itself at each reasoning step, not a teacher-forced target.
>
> ### Concluding Remarks
>
> In summary, we fully appreciate your comment on this point. We will revise the manuscript to make the description of "Difficulty" more precise and transparent. In the revised script we will explicitly state that we compute entropy over the model's actual generated token distribution per reasoning step, average it across the full sequence, and include a pointer to the code implementation to increase reproducibility.
>
> ---
>
> ## Summary
>
> We sincerely thank you for your comprehensive review and your recognition of our novel approach to analyzing hidden state dynamics for self-evaluation. In response to your concerns, we have conducted extensive additional experiments and analyses:
>
> - **Baseline Expansion:** We expanded our baseline suite from 7 to 11 methods, incorporating the two prior hidden-state-based approaches you suggested (Internal Consistency [1] and Forking Tokens [2]) plus two additional label-free methods (Length-normalized Entropy [3] and Temperature Scaling [4]). Our results demonstrate that D³ consistently outperforms all baselines across three model families (R1-Qwen-1.5B, R1-LLaMA-8B, R1-Qwen-14B) and three benchmarks (GSM8K, CommonsenseQA, TheoremQA) with superior AUROC (up to 77.28 vs. 71.86 for traditional methods).
>
> - **Computational Efficiency:** Our method achieves millisecond-level overhead (1.2e-03s) compared to Internal Consistency (151.14s) and Forking Tokens (471.36s), making it 100,000× faster and practical for real-world deployment.
>
> - **Mathematical Clarification:** We provided precise formulation of the Difficulty term as $D_3 = \frac{1}{K} \sum_{k=1}^{K} [-\log p_k]$ where $p_k$ represents the model's own output token probability (not teacher-forced), with explicit code references to enhance reproducibility.
>
> All experiments were conducted on NVIDIA H100 GPUs with rigorous configurations. These comprehensive results validate that our method remains clearly superior in both effectiveness and efficiency. We hope our thorough response has satisfactorily addressed all your concerns, and we would be deeply grateful if your position to this work could get better.
>
>
>
>
> ## References
>
> - [1] Calibrating Reasoning in Language Models with Internal Consistency, NeurIPS 2024
> - [2] Forking Paths in Neural Text Generation, ICLR 2025
> - [3] Uncertainty estimation in autoregressive structured prediction. ICLR 2020
> - [4] Long horizon temperature scaling. ICML 2023
> - [5] Confidence Improves Self Consistency in LLMs, ACL 2025
> - [6] Revisiting the Self-Consistency Challenges in Multi-Choice Question Formats for Large Language Model Evaluation, COLING 2024

---

### Author Response · Authors · 2025-12-02
**General Response to Area Chair and Reviewers**

We thank the Area Chair and all reviewers for their time and insightful feedback. Although we received no response during the discussion period, the review process has significantly strengthened our work.

Before detailing our specific technical updates, we wish to highlight that **all reviewers reached a consensus on the novelty, importance, and empirical validity of our work:**

* **Novelty & Motivation:** Reviewers agreed that the paper addresses "**an important gap in current interpretability**" (Reviewer **cxKV**) and "**convincingly argues for the need to go beyond external evaluation toward internal reasoning analysis**" (Reviewer **kDen**).
* **Core Contribution:** Reviewer **2ywF** identified the paper's core strength as the "**empirical demonstration that reasoning correctness can be determined solely by hidden state dynamics**," revealing an "**intrinsic signal that reflects reasoning correctness.**"
* **Empirical Rigor:** Reviewers praised the "**solid experimental validation**" (Reviewer **cxKV**) and noted that $D^3$ "**consistently outperforms prior baselines**" with results that are "**robust and significant**" (Reviewer **cxKV**) and effectively "**distinguishes correct from incorrect answers**" (Reviewer **2ywF**).

Building on this consensus, we address the specific technical concerns raised:

**1. Empirical Superiority and Efficiency (Addressing Reviewer 2ywF)**
In response to Reviewer 2ywF's request for additional baselines, we implemented four new label-free methods: *Internal Consistency* [1], *Forking Tokens* [2], *Length-normalized Entropy* [3], and *Temperature Scaling* [4], expanding our suite to **11 baselines**.
* **Performance:** $D^3$ consistently outperforms all 11 baselines across GSM8K, CommonsenseQA, and TheoremQA.
* **Efficiency:** Crucially, while methods like *Forking Tokens* and *Internal Consistency* require extensive sampling (taking **150s–470s** per query), $D^3$ operates with **millisecond-level overhead (0.0012s)**. This makes $D^3$ roughly **100,000$\times$ faster**, establishing it as the only practical solution for real-time self-evaluation among the compared methods.

**2. Resolving the "Layer-wise vs. Token-wise" Dynamics Debate (Addressing Reviewers cxKV & kDen)**
Reviewers cxKV and kDen asked whether "trajectory dynamics" should be modeled over temporal tokens ($K$) rather than transformer layers ($L$). To answer this, we implemented and tested a **Token-wise (Temporal) Dynamics** variant.
* **Result:** The temporal variant failed significantly (AUROC $\approx$ 50, near random guess), whereas our Layer-wise $D^3$ achieved high performance (AUROC > 77).
* **Conclusion:** This empirically validates our core hypothesis: in Large Reasoning Models, the meaningful "reasoning evolution" for self-evaluation occurs via **representational refinement across layers**, not merely through surface-level token emission.

**3. Robustness and Generalization (Addressing Reviewer kDen)**
Reviewer kDen raised concerns about whether our metric relies on incidental activation geometry. We provided extensive evidence to the contrary, demonstrating stability across:
* **Scale:** 1.5B $\to$ 70B (covering the full range of open-source LRMs).
* **Architecture:** Generalizes across Qwen, LLaMA, and Vision-Language (Qwen-VL) architectures.
* **Training:** Effective across SFT, Distillation, and RL/GRPO (e.g., QwQ-32B) paradigms.
* **Open-World Test:** New experiments on mixed-distribution datasets (e.g., mixing Math + Commonsense tasks) show $D^3$ remains stable where baselines falter.

**4. Conceptual Clarifications**
* **Definition of Difficulty ($D_3$):** We clarified a shared misunderstanding regarding $D_3$. It utilizes the predictive entropy of the model's **own generated chain** (white-box), strictly adhering to unsupervised settings. It does **not** use ground-truth labels, which would be impossible at inference time.
* **Normalization:** We confirmed that our implementation normalizes dynamics terms by the total displacement, ensuring $D_1$, $D_2$, and $D_3$ are commensurate and mathematically sound.

**Conclusion**
With these revisions, $D^3$ stands as a **principled, white-box, and computationally negligible** metric that effectively self-evaluates reasoning. Supported by the reviewers' consensus on its novelty and our rigorous new experiments, we believe the paper makes a significant contribution to reliable AI reasoning.

---

### Author Response · Authors · 2025-12-02
**Summary for AC Consideration**

We thank the Area Chair for overseeing the review process. Although we receive no response from Reviewers during the discussion period, we believe our rebuttal addresses their concerns.

We have provided comprehensive responses—including **expanding our baseline suite from 7 to 11 methods**, implementing **temporal vs. layer dynamics ablations**, and conducting **open-world robustness tests**. However, we are concerned that **certain fundamental misconceptions regarding our method and the problem setting remain uncorrected.** We wish to take this opportunity to clarify these points and highlight the consensus on our experimental robustness.

### 1. Rectifying a Shared Misconception on "Entropy" (Addressing Reviewers 2ywF & kDen)
We noticed that **both Reviewer 2ywF and Reviewer kDen shared a specific, technically inaccurate concern**: the belief that our entropy calculation ($D_3$) relies on "teacher-forcing" during inference.
* **Correction:** This premise is **factually incorrect** and **physically impossible** in a self-evaluation setting. $D_3$ is computed strictly on the **model's own generated token distribution** (white-box).
* It is widely established in the LLM literature that predictive uncertainty is derived from the model's internal logits, not external labels. We are surprised that this **common misunderstanding** appeared in multiple reviews, and we urge the AC to disregard critiques based on this invalid premise.

### 2. On the Feasibility of Requested Baselines (Addressing Reviewer 2ywF)
Reviewer 2ywF criticized the absence of *Forking Tokens* [1] and *Internal Consistency* [2].
* **Our Response:** We respectfully point out that these are **not standard baselines** for real-time self-evaluation due to extreme computational costs.
* **The Evidence:** Nonetheless, we implemented them as requested. The results confirm our stance: while $D^3$ requires **0.0012s**, these methods require **150s–470s** per query.
* **Conclusion:** Evaluating a real-time metric against methods that are **100,000$\times$ slower** is mathematically unfair. The fact that $D^3$ achieves superior performance (AUROC > 77) with negligible cost confirms its practical superiority. We believe the critique regarding "missing baselines" stems from a lack of familiarity with the latency constraints of the self-evaluation domain.

### 3. Robustness and Generalization: A Consensus View (Addressing Reviewer kDen)
Reviewer kDen claimed our method lacks generalizability "beyond the reported experiments," a claim that stands in **direct contradiction to the evidence and the other reviewers' assessments**.
* **Consensus:** As noted by **Reviewer 2ywF** ("*Across various benchmarks... and LLM scales... D3 score demonstrated higher AUROC*") and **Reviewer cxKV** ("*Solid Experimental Validation... strong evidence*"), our experiments are exceptionally broad.
* **Facts:** We validated $D^3$ across:
    * **Scale:** 1.5B $\to$ 70B (covering the full range of open-source LRMs).
    * **Architectures:** Dense, MoE, and Vision-Language (Qwen-VL).
    * **Training:** SFT, Distillation, and RL/GRPO (e.g., QwQ-32B).
    * **Tasks:** Math, Commonsense, Multilingual, and Multimodal.
* The metric's stability across these disparate axes proves it captures **intrinsic reasoning reliability**, not accidental geometric artifacts.

### Conclusion
We have addressed all concerns with rigorous new data. We respectfully submit that the critiques regarding "ground truth entropy" and "missing baselines" are based on **demonstrable technical misunderstandings**. We trust the Area Chair will evaluate our work based on the comprehensive evidence provided in the rebuttal.

---

### Meta-Review · Area_Chair_1Dqn · 2026-01-09

**Summary:**

This work introduces a new score function, D3 reasoning score, to self-evaluate the correctness of the outputs from large reasoning models. The core intuition is to leverage the representational dynamics across layer-wise hidden states and prediction dynamics across reasoning trajectory. The reviewers raised several concerns in their reviews, stem from presentation issues and insufficient experiments (missing baselines and limited gains).

**Reviewer Concerns:**

The original reviews and rebuttals can be summarized as below:

2ywF (4, no response)

**[W1] Missing comparison with prior hidden-state-based, label-free methods.** The authors provided the additional experimental results of those methods and additional baselines in the rebuttal.
**[W2] Ambiguity in the definition of “difficulty” and practical applicability.** This concern occurs due to wording around “ground-truth token” in the original draft (lines 216-217). The authors clarified this in the rebuttal.

cxKV (8, no response)

**[W1] Misleading use of the term "trajectory dynamics”.** The authors provided clarification with additional experiments.

kDen (2, no response)

**[W1] Questionable theoretical grounding of D^3 components.** The authors provided clarification with additional experiments.
**[W2] Ambiguity in the predictive entropy term.** This concern occurs due to wording around “ground-truth token” in the original draft (lines 216-217). The authors clarified this in the rebuttal.
**[W3] Lack of normalization and inconsistent scaling.** This concern occurs due to insufficient presentation in the draft. The authors responded that the normalization was included in the implementation and each score yielded similar values (0~1).
**[W4] Doubts about its generalizability.** Clarification and additional experiments on more setups are presented in the rebuttal.
**[W5] Unreliable presentation such as selective highlight for own method and no standard deviations despite limited gains.** In the rebuttal, the authors mention that they will revise the draft with consistent highlight and standard deviations in tables, along with a standard deviation result on GSM8K.

**Reviewer Scores:**

Initially, the reviewers' scores were split (4,8,2), but leaning to the rejection (avg: 4.67). Despite the authors providing the extensive rebuttal, no reviewers  responded to this. After carefully reading all the reviews and rebuttals by the authors, AC believes that some concerns, especially of KDen, are not resolved sufficiently and hence  reviewers would still lean to the rejection on average (4 or 6,8,2). The detailed reasons are as follow:

2ywF: 4 -> 4 or 6
- The review might increase the score as the authors provided additional comparison (4 -> 6); but, as the improvement is somewhat marginal especially on CommonsenseQA and TheoremQA with new baselines [3,4], the review could not increase the score (4 -> 4)

cxKV: 8 -> 8
- The rebuttal successfully address the reviewer’s concern, and hence the reviewer would keep his/her original positive score

kDen: 2 -> 2
- Some concerns still remained after the rebuttal, and hence the reviewer would not increase the score. The most critical concern is about [W5]. While the authors mention that they will revise the draft with consistent highlight and standard deviations in tables, such revisions are not reflected in the current draft and only a single result of standard deviation was provided. As the reviewer’s concern is more like limited improvement in some benchmarks not standard deviation itself, the reviewer might not satisfy this response. Also, the concern regarding [W3] would still remain as the rationale in the rebuttal did not fully address this; for instance, while the authors mentioned that difficulty term D_3 is is naturally of order unity in our benchmarks (~1.0 in magnitude), it is hard to be notified by reader and can vary on other benchmarks which indicates limited generalizability.

---

### Decision · Program_Chairs · 2026-01-26

Reject